# DATA LEAKAGE IN TABULAR FEDERATED LEARNING

## ABSTRACT

While federated learning (FL) promises to preserve privacy in distributed training of deep learning models, recent work in the image and NLP domains showed that training updates leak private data of participating clients. At the same time, most high-stakes applications of FL (e.g., legal and financial) use tabular data. Compared to the NLP and image domains, reconstruction of tabular data poses several unique challenges: (i) categorical features introduce a significantly more difficult mixed discrete-continuous optimization problem, (ii) the mix of categorical and continuous features causes high variance in the final reconstructions, and (iii) structured data makes it difficult for the adversary to judge reconstruction quality. In this work, we tackle these challenges and propose the first comprehensive reconstruction attack on tabular data, called TabLeak. TabLeak is based on three key ingredients: (i) a softmax structural prior, implicitly converting the mixed discrete-continuous optimization problem into an easier fully continuous one, (ii) a way to reduce the variance of our reconstructions through a pooled ensembling scheme exploiting the structure of tabular data, and (iii) an entropy measure which can successfully assess reconstruction quality. Our experimental evaluation demonstrates the effectiveness of TabLeak, reaching a state-of-the-art on four popular tabular datasets. For instance, on the Adult dataset, we improve attack accuracy by 10% compared to the baseline on the practically relevant batch size of 32 and further obtain non-trivial reconstructions for batch sizes as large as 128. Our findings are important as they show that performing FL on tabular data, which often poses high privacy risks, is highly vulnerable.

## 1 INTRODUCTION

Federated Learning (McMahan et al., 2016) (FL) has emerged as the most prominent approach to training machine learning models collaboratively without requiring sensitive data of different parties to be sent to a single centralized location. While prior work has examined privacy leakage in federated learning in the context of computer vision (Zhu et al., 2019; Geiping et al., 2020; Yin et al., 2021) and natural language processing (Dimitrov et al., 2022a; Gupta et al., 2022; Deng et al., 2021), many applications of FL rely on large tabular datasets that include highly sensitive personal data such as financial information and health status (Borisov et al., 2021; Rieke et al., 2020; Long et al., 2021). However, no prior work has studied the issue of privacy leakage in the context of tabular data, a cause of concern for public institutions which have recently launched a competition[1] with a 1.6 mil. USD prize to develop privacy-preserving FL solutions for fraud detection and infection risk prediction, both being tabular datasets.

**Key challenges** Leakage attacks often rely on solving optimization problems whose solutions are the desired sensitive data points. Unlike other data types, tabular data poses unique challenges to solving these problems because: (i) the reconstruction is a solution to a mixed discrete-continuous optimization problem, in contrast to other domains where the problem is either fully continuous or discrete (pixels for images and tokens for text), (ii) there is high variance in the final reconstructions because, uniquely to tabular data, discrete changes in the categorical features significantly change the optimization trajectory, and (iii) assessing the quality of reconstructions is harder compared to images and text - e.g. determining whether a person with given reconstructed characteristics exists is difficult. Together, these challenges imply that it is difficult to make existing attacks work on tabular data.

---

[1]https://petsprizechallenges.com/

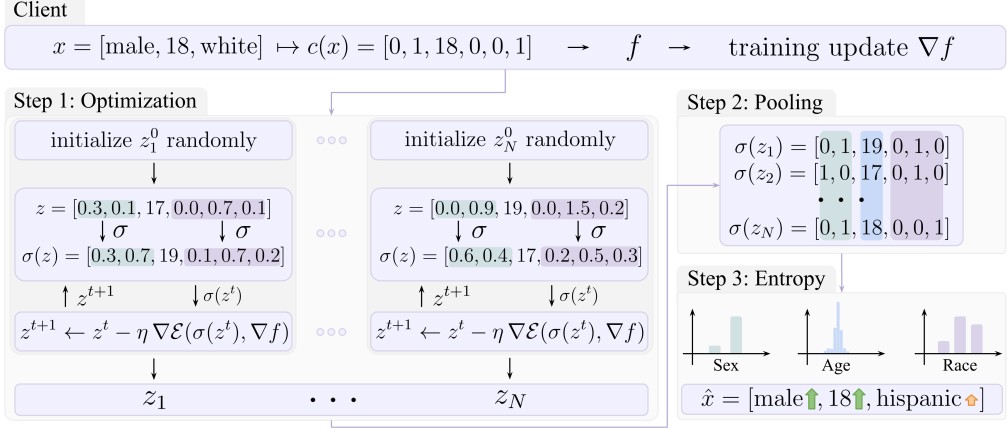

Figure 1: Overview of TabLeak. Our approach transforms the optimization problem into a fully continuous one by optimizing continuous versions of the discrete features, obtained by applying softmax (Step 1, middle boxes), resulting in $N$ candidate solutions (Step 1, bottom). Then, we pool together an ensemble of $N$ different solutions $z_1, z_2, ..., z_N$ obtained from the optimization to reduce the variance of the reconstruction (Step 2). Finally, we assess the quality of the reconstruction by computing the entropy from the feature distributions in the ensemble (Step 3).

**This work**  In this work, we propose the first comprehensive leakage attack on tabular data in the FL setting, addressing the previously mentioned challenges. We provide an overview of our approach in Fig. 1, showing the reconstruction of a client's private training data point $x = [\text{male}, 18, \text{white}]$, from the corresponding training update $\nabla f$ received by the server. In Step 1, we create $N$ separate optimization problems, each assigning different initial values $z_1^0, \ldots z_N^0$ to the optimization variables, representing our reconstruction of the client's one-hot encoded data, $c(x)$. To address the first challenge of tabular data leakage, we transform the mixed discrete-continuous optimization problem into a fully continuous one, by passing our current reconstructions $z_1^t, \ldots, z_N^t$ through a per-feature softmax $\sigma$ at every step $t$. Using the softmaxed data $\sigma(z^t)$, we take a gradient step to minimize the reconstruction loss, which compares the received client update $\nabla f$ with a simulated client update computed on $\sigma(z^t)$. In Step 2, we reduce the variance of the final reconstruction by performing pooling over the $N$ different solutions $z_1, z_2, ..., z_N$, thus tackling the second challenge. In Step 3, we address the challenge of assessing the fidelity of our reconstructions. We rely on the observation that often when our proposed reconstructions $z_1, z_2, ..., z_N$ agree they also match the true client data, $c(x)$. We measure the agreement using entropy. In the example above, we see that the features *sex* and *age* produced a low entropy distribution. Therefore we assign high confidence to these results (green arrows). In contrast, the reconstruction of the feature *race* receives a low confidence rating (orange arrow); rightfully so, as the reconstruction is incorrect.

We implemented our approach in an end-to-end attack called TabLeak and evaluated it on several tabular datasets. Our attack is highly effective: it can obtain non-trivial reconstructions for batch sizes as large as 128, and on many practically relevant batch sizes such as 32, it improved reconstruction accuracy by up to 10% compared to the baseline. Overall, our findings show that FL is highly vulnerable when applied to tabular data.

**Main contributions**  Our main contributions are:

- Novel insights enabling efficient attacks on FL with tabular data: using softmax to make the optimization problem fully continuous, ensembling to reduce the variance, and entropy to assess the reconstructions.

- An implementation of our approach into an end-to-end tool called TabLeak.

- Extensive experimental evaluation, demonstrating effectiveness of TabLeak at reconstructing sensitive client data on several popular tabular datasets.

## 2 BACKGROUND AND RELATED WORK

In this section, we provide the necessary technical background for our work, introduce the notation used throughout the paper, and present the related work in this field.

**Federated Learning**  Federated Learning (FL) is a training protocol developed to facilitate the distributed training of a parametric model while preserving the privacy of the data at source (McMahan et al., 2016). Formally, we have a parametric function $f_\theta$, where $\theta \in \Theta$ are the (network) parameters and $f_\theta : \mathcal{X} \to \mathcal{Y}$. Given a dataset as the union of private datasets of clients $\mathcal{S} = \bigcup_{k=1}^{K} \mathcal{S}_k$, we now wish to find a $\theta^* \in \Theta$ such that:

$$\theta^* = \arg\min_{\theta \in \Theta} \; \frac{1}{N} \sum_{(x_i, y_i) \in \mathcal{S}} \mathcal{L}(f_\theta(x_i), y_i), \tag{1}$$

in a distributed manner, *i.e.* without collecting the dataset $\mathcal{S}$ in a central database. McMahan et al. (2016) propose two training algorithms: FedSGD (a similar algorithm was also proposed by Shokri & Shmatikov (2015)) and FedAvg, that allow for the distributed training of $f_\theta$, while keeping the data partitions $\mathcal{S}_k$ at client sources. The two protocols differ in how the clients compute their local updates in each step of training. In FedSGD, each client calculates the update gradient with respect to a randomly selected batch of their own data and shares the resulting gradient with the server. In FedAvg, the clients conduct a few epochs of local training on their own data before sharing their resulting parameters with the server. After the server has received the gradients/parameters from the clients, it aggregates them, updates the model, and broadcasts it to the clients. In each case, this process is repeated until convergence, where FedAvg usually requires fewer rounds of communication.

**Data Leakage Attacks**  Although FL was designed with the goal of preserving the privacy of clients' data, recent work has uncovered substantial vulnerabilities. Melis et al. (2019) first presented how one can infer certain properties of the clients' private data in FL. Later, Zhu et al. (2019) demonstrated that an *honest but curious* server can use the current state of the model and the client gradients to reconstruct the clients' data, breaking the main privacy promise of Federated Learning (FL). Under this threat model, there has been extensive research on designing tailored attacks for images (Geiping et al., 2020; Geng et al., 2021; Huang et al., 2021; Jin et al., 2021; Balunović et al., 2021; Yin et al., 2021; Zhao et al., 2020; Jeon et al., 2021; Dimitrov et al., 2022b) and natural language (Deng et al., 2021; Dimitrov et al., 2022a; Gupta et al., 2022). However, no prior work has comprehensively dealt with tabular data, despite its significance in real-world high-stakes applications (Borisov et al., 2021). Some works also consider a threat scenario where the malicious server is allowed to change the model or the updates communicated to the clients (Wen et al., 2022; Fowl et al., 2021); but in this work we focus on the honest-but-curious setting.

In training with FedSGD, given the model $f_\theta$ at an iteration $t$ and the gradient $\nabla_\theta \mathcal{L}(f_\theta(x), y)$ of some client, we solve the following optimization problem to retrieve the client's private data:

$$\hat{x}, \hat{y} = \arg\min_{(x', y') \in \mathcal{X} \times \mathcal{Y}} \mathcal{E}(\nabla_\theta \mathcal{L}(f_\theta(x), y), \nabla_\theta \mathcal{L}(f_\theta(x'), y')) + \lambda \mathcal{R}(x'). \tag{2}$$

Where in Eq. (2) we denote the *gradient matching loss* as $\mathcal{E}$ and $\mathcal{R}$ is an optional regularizer for the reconstruction. The work of Zhu et al. (2019) used the mean square error for $\mathcal{E}$, on which Geiping et al. (2020) improved using the cosine similarity loss. Zhao et al. (2020) first demonstrated that the private labels $y$ can be estimated before solving Eq. (2), reducing the complexity of Eq. (2) and improving the attack results. Their method was later extended to batches by Yin et al. (2021) and refined by Geng et al. (2021). Eq. (2) is typically solved using continuous optimization tools such as L-BFGS (Liu & Nocedal, 1989) and Adam (Kingma & Ba, 2014). Although analytical approaches exist, they do not generalize to batches with more than a single data point (Zhu & Blaschko, 2020).

Depending on the data domain, distinct tailored alterations to Eq. (2) have been proposed in the literature, *e.g.,* using the total variation regularizer for images (Geiping et al., 2020) and exploiting pre-trained language models in language tasks (Dimitrov et al., 2022a; Gupta et al., 2022). These mostly non-transferable domain-specific solutions are necessary as each domain poses unique challenges. Our work is first to identify and tackle the key challenges to data leakage in the tabular domain.

**Mixed Type Tabular Data** Mixed type tabular data is a data type commonly used in health, economic and social sciences, which entail high-stakes privacy-critical applications (Borisov et al., 2021). Here, data is collected in a table of feature columns, mostly human-interpretable, *e.g.,* age, nationality, and occupation of an individual. We formalize tabular data as follows. Let $x \in \mathcal{X}$ be one *line* of data, containing discrete or *categorical* features and continuous or *numerical* features. Let $\mathcal{X}$ contain $K$ discrete feature columns and $L$ continuous feature columns, *i.e.* $\mathcal{X} = \mathcal{D}_1 \times \mathcal{D}_2 \times \cdots \times \mathcal{D}_K \times \mathcal{U}_1 \times \cdots \times \mathcal{U}_L$, where $\mathcal{D}_i \subset \mathbb{N}$ with cardinality $|\mathcal{D}_i| = D_i$ and $\mathcal{U}_i \subset \mathbb{R}$. For the purpose of deep neural network training, the categorical features are often encoded in a numerical vector. We denote the encoded data batch or line as $c(x)$, where we preserve the continuous features and encode the categorical features by a *one-hot* encoding. The one-hot encoding of the $i$-th discrete feature $x_i^D$ is a vector $c_i^D(x)$ of length $D_i$ that has a one at the position marking the encoded category, while all other entries are zeros. We obtain the represented category by taking the argmax of $c_i^D(x)$ (projection to obtain $x$). Using the described encoding, one line of data $x \in \mathcal{X}$ translates to: $c(x) = \left[c_1^D(x), c_2^D(x), \ldots, c_K^D(x), x_1^C, \ldots, x_L^C\right]$, containing $d := L + \sum_{i=1}^{K} D_i$ entries.

## 3 TABULAR LEAKAGE

In this section, we briefly summarize the key challenges in tabular leakage and present our solution to these challenges in the subsequent subsections and our end-to-end attack.

**Key Challenges** We now list the three key challenges that we address in our work: (i) the presence of both categorical and continuous features in tabular data require the attacker to solve a significantly harder mixed discrete-continuous optimization problem (addressed in Sec. 3.1), (ii) the large distance in the encodings of the categorical features introduces high variance in the leakage problem (addressed in Sec. 3.2), and (iii) in contrast to images and text, it is hard for an adversary to assess the quality of the reconstructed data in the tabular domain, as most reconstructions may be projected to credible input data points (we address this via an uncertainty quantification scheme in Sec. 3.3).

### 3.1 THE SOFTMAX STRUCTURAL PRIOR

We now discuss our solution to challenge (i) – we introduce the softmax structural prior, which turns the hard mixed discrete-continuous optimization problem into a fully continuous one. This drastically reduces its complexity, while still facilitating the recovery of correct discrete structures.

To start, notice that the recovery of one-hot encodings can be enforced by ensuring that all entries of the recovered vector are either zero or one, and exactly one of the entries equals to one. However, these constraints enforce integer properties, *i.e.* they are non-differentiable and can not be used in combination with the powerful continuous optimization tools used for gradient leakage attacks. Relaxing the integer constraint by allowing the reconstructed entries to take real values in $[0, 1]$, we are still left with a constrained optimization problem not well suited for popular continuous optimization tools, such as Adam (Kingma & Ba, 2014). Therefore, we are looking for a method that can implicitly enforce the constraints introduced above.

Let $z \in \mathbb{R}^d$ be our approximate intermediate solution for the true one-hot encoded data $c(x)$ at some optimization step. Then, we are looking for a differentiable function $\sigma : \mathbb{R}^{D_i} \to [0, 1]$, such that:

$$\sum_{j=1}^{D_i} \sigma(z_i^D)[j] = 1 \qquad \text{and} \qquad \sigma(z_i^D)[j] \in [0, 1] \quad \forall j \in \mathcal{D}_i. \tag{3}$$

Notice that the two conditions in Eq. (3) can be fulfilled by applying a *softmax* to $z_i^D$, *i.e.* define:

$$\sigma(z_i^D)[j] := \frac{\exp(z_i^D[j])}{\sum_{k=1}^{D_i} \exp(z_i^D[k])} \qquad \forall j \in \mathcal{D}_i. \tag{4}$$

Note that it is easy to show that Eq. (4) fulfills both conditions in Eq. (3) and that it is differentiable. Putting this together, in each round of optimization we will have the following approximation of the true data point: $c(x) \approx \sigma(z) = \left[\sigma(z_1^D), \ldots, \sigma(z_K^D), z_1^C, \ldots, z_L^C\right]$. In order to preserve notational simplicity, we write $\sigma(z)$ to mean the application of softmax to each group of entries representing a given categorical variable separately.

## 3.2 Pooled Ensembling

As mentioned earlier, the mix of categorical and continuous features introduces further variance in the difficult reconstruction problem which already has multiple local minima and high sensitivity to initialization (Zhu & Blaschko, 2020) (challenge (ii)). Concretely, as the one-hot encodings of the categorical features are orthogonal to each other, a change in the encoded category can drastically change the optimization trajectory. We alleviate this problem by adapting an established method of variance reduction in noisy processes (Hastie et al., 2009), *i.e.* we run independent optimization processes with different initializations and ensemble their results through feature-wise pooling.

Note that the features in tabular data are tied to a certain position in the recovered data vector, thereby we can combine independent reconstructions to obtain an improved and more robust final estimate of the true data by applying feature-wise pooling. Formally, we run $N$ independent rounds of optimization with $i.i.d.$ initializations recovering potentially different reconstructions $\{\sigma(z_j)\}_{j=1}^N$. Then, we obtain a final estimate of the true encoded data, denoted as $\sigma_i^D(\hat{z})$, by pooling them:

$$\sigma_i^D(\hat{z}) = \text{pool}\left(\left\{\sigma(z_{ji}^D)\right\}_{j=1}^N\right) \quad \forall i \in [K] \quad \text{and} \quad \hat{z}_i^C = \text{pool}\left(\left\{(z_{ji}^C)\right\}_{j=1}^N\right) \quad \forall i \in [L]. \quad (5)$$

Where the pool$(\cdot)$ operation can be any permutation invariant mapping that maps to the same structure as its inputs. In our attack, we use median pooling for both continuous and categorical features.

Notice that because a batch-gradient is invariant to permutations of the datapoints in the corresponding batch, when reconstructing from such a gradient we may retrieve the batch-points in a different order at every optimization instance. Hence, we need to reorder each batch such that their lines match to each other, and only then we can conduct the pooling. We reorder by first selecting the sample that produced the best reconstruction loss at the end of optimization $\hat{z}^{best}$, with projection $\hat{x}^{best}$. Then, we match the lines of every other sample in the collection with respect

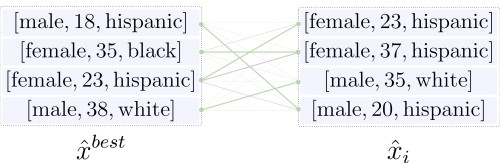

Figure 2: Maximum similarity matching of a sample from the collection of reconstructions to the best-loss sample $\hat{x}^{best}$.

to $\hat{x}^{best}$. Concretely, we calculate the similarity (described in detail in Sec. 4) between each pair of lines of $\hat{x}^{best}$ and another sample $\hat{x}_i$ in the collection and find the maximum similarity reordering of the lines with the help of bipartite matching solved by the Hungarian algorithm (Kuhn, 1955). This process is depicted in Fig. 2. Repeating this for each sample, we reorder the entire collection with respect to the best-loss sample, effectively reversing the permutation differences in the independent reconstructions. Therefore, after this process we can directly apply feature-wise pooling for each line over the collection.

## 3.3 Entropy-based Uncertainty Estimation

We now address challenge (iii) above. To recap, it is significantly harder for an adversary to assess the quality of an obtained reconstruction when it comes to tabular data, as almost any reconstruction may constitute a credible data point when projected back to mixed discrete-continuous space. Note that this challenge does not arise as prominently in the image (or text) domain, because by looking at a picture one can easily judge if it is just noise or an actual image. To address this issue, we propose to estimate the reconstruction uncertainty by looking at the level of agreement over a certain feature for different reconstructions. Concretely, given a collection of reconstructions as in Sec. 3.2, we can observe the distribution of each feature over the reconstructions. Intuitively, if this distribution is "peaky", *i.e.* concentrates the mass heavily on a certain value, then we can assume that the feature has been reconstructed correctly, whereas if there is high disagreement between the reconstructed samples, we can assume that this feature's recovered final value should not be trusted. We can quantify this by measuring the entropy of the feature distributions induced by the recovered samples.

**Categorical Features** Let $p(\hat{x}_i^D)_m := \frac{1}{N} \text{Count}_j(\hat{x}_{ji}^D = m)$ be the relative frequency of projected reconstructions of the $i$-th discrete feature of value $m$ in the ensemble. Then, we can calculate the normalized entropy of the feature as $\bar{H}_i^D = \frac{-1}{\log |\mathcal{D}_i|} \sum_{m=1}^{D_i} p(\hat{x}_i^D)_m \log p(\hat{x}_i^D)_m$. Note that the normalization allows for comparing features with supports of different size, *i.e.* it ensures that $\bar{H}_i^D \in [0, 1]$, as $0 \leq H(k) \leq \log |\mathcal{K}|$ for any discrete random variable $k \in \mathcal{K}$ of finite support.

---

**Algorithm 1** Our combined attack against training by FedSGD

---

1: **function** SINGLEINVERSION(Neural Network: $f_\theta$, Client Gradient: $\nabla_\theta \mathcal{L}(f_\theta(c(x)), y)$, Reconstructed Labels: $\hat{y}$, Initial Reconstruction: $z_i^0$, Iterations: $T$, N. of Discrete Features: $K$)
2:     **for** $t$ **in** $1, 2, \ldots, T$ **do**
3:         **for** $k$ **in** $1, 2, \ldots, K$ **do**
4:             $\sigma(z_{ik}^D) \leftarrow$ SOFTMAX$(z_{ik}^D)$
5:         **end for**
6:         $z_i^{t+1} \leftarrow z_i^t - \eta \, \nabla_z \mathcal{E}_{CS}(\nabla_\theta \mathcal{L}(f_\theta(c(x)), y), \nabla_\theta \mathcal{L}(f_\theta(\sigma(z_i^t)), \hat{y}))$
7:     **end for**
8:     **return** $z_i^T$
9: **end function**
10:
11: **function** TABLEAK(Neural Network: $f_\theta$, Client Gradient: $\nabla_\theta \mathcal{L}(f_\theta(c(x)), y)$, Reconstructed Labels: $\hat{y}$, Ensemble Size: $N$, Iterations: $T$, N. of Discrete Features: $K$)
12:     $\left\{z_i^0\right\}_{i=1}^N \sim \mathcal{U}_{[0,1]^d}$
13:     **for** $i$ **in** $1, 2, \ldots, N$ **do**
14:         $z_i^T \leftarrow$ SINGLEINVERSION$(f_\theta, \nabla_\theta \mathcal{L}(f_\theta(c(x)), y), \hat{y}, z_i^0, T, K)$
15:     **end for**
16:     $\hat{z}^{best} \leftarrow \arg\min_{z_j^T} \mathcal{E}_{CS}(\nabla_\theta \mathcal{L}(f_\theta(c(x)), y), \nabla_\theta \mathcal{L}(f_\theta(\sigma(z_j^T)), \hat{y}))$
17:     $\sigma(\hat{z}) \leftarrow$ MATCHANDPOOL$(\left\{\sigma(z_i^T)\right\}_{i=1}^N, \sigma(\hat{z}^{best}))$
18:     $\bar{H}^D, H^C \leftarrow$ CALCULATEENTROPY$(\left\{\sigma(z_i^T)\right\}_{i=1}^N)$
19:     $\hat{x} \leftarrow$ PROJECT$(\sigma(\hat{z}))$
20:     **return** $\hat{x}, \bar{H}^D, H^C$
21: **end function**

---

**Continuous Features** In case of the continuous features, we calculate the entropy assuming that errors of the reconstructed samples follow a Gaussian distribution. As such, we first estimate the sample variance $\hat{\sigma}_i^2$ for the $i$-th continuous feature and then plug it in to calculate the entropy of the corresponding Gaussian: $H_i^C = \frac{1}{2} + \frac{1}{2} \log 2\pi\hat{\sigma}_i^2$. As this approach is universal over all continuous features, it is enough to simply scale the features themselves to make their entropy comparable. For example, this can be achieved by working only with standardized features.

Note that as the categorical and the continuous features are fundamentally different from an information theoretic perspective, we have no robust means to combine them in a way that would allow for equal treatment. Therefore, when assessing the credibility of recovered features, we will always distinguish between categorical and continuous features.

### 3.4 COMBINED ATTACK

Now we provide the description of our end-to-end attack, TabLeak. Following Geiping et al. (2020), we use the cosine similarity loss as our reconstruction loss, defined as:

$$\mathcal{E}_{CS}(\nabla_{\theta_t} \mathcal{L}(f_{\theta_t}(c(x)), y), \nabla_{\theta_t} \mathcal{L}(f_{\theta_t}(\sigma(z)), \hat{y})), \quad \text{with} \quad \mathcal{E}_{CS}(l, g) := 1 - \frac{\langle l, g \rangle}{\|l\|_2 \, \|g\|_2}, \quad (6)$$

where $(x, y)$ are the true data, $\hat{y}$ are the labels reconstructed beforehand, and we optimize for $z$. Our algorithm is shown in Alg. 1. First, we reconstruct the labels using the label reconstruction method of Geng et al. (2021) and provide them as an input to our attack. Then, we initialize $N$ independent dummy samples for an ensemble of size $N$ (Line 12). Starting from each initial sample we optimize independently (Line 13-15) via the SINGLEINVERSION function. In each optimization step, we apply the softmax structural prior of Sec. 3.1, and let the optimizer differentiate through it (Line 4). After the optimization processes have converged or have reached the maximum number of allowed iterations $T$, we identify the sample $\hat{z}^{best}$ producing the best reconstruction loss (Line 16). Using this sample, we match and median pool to obtain the final encoded reconstruction $\sigma(\hat{z})$ in Line 17 as described in Sec. 3.2. Finally, we return the projected reconstruction $\hat{x}$ and the corresponding feature-entropies $\bar{H}^D$ and $H^C$, quantifying the uncertainty in the reconstruction.

## 4 EXPERIMENTAL EVALUATION

In this section, we first detail the evaluation metric we used to assess the obtained reconstructions. We then briefly explain our experimental setup. Next, we evaluate our attack in various settings against baseline methods, establishing a new state-of-the-art. Finally, we demonstrate the effectiveness of our entropy-based uncertainty quantification method.

**Evaluation Metric**  As no prior work on tabular data leakage exists, we propose our metric for measuring the accuracy of tabular reconstruction, inspired by the 0-1 loss, allowing the joint treatment of categorical and continuous features. For a reconstruction $\hat{x}$, we define the accuracy metric as:

$$\text{accuracy}(x, \hat{x}) \coloneqq \frac{1}{K + L} \left( \sum_{i=1}^{K} \mathbb{I}\{x_i^D = \hat{x}_i^D\} + \sum_{i=1}^{L} \mathbb{I}\{\hat{x}_i^C \in [x_i^C - \epsilon_i, \ x_i^C + \epsilon_i]\} \right), \quad (7)$$

where $x$ is the ground truth and $\{\epsilon_i\}_{i=1}^{L}$ are constants determining how close the reconstructed continuous features have to be to the original value in order to be considered successfully leaked. We provide more details on our metric in App. A and experiments with additional metrics in App. C.3.

**Baselines**  We consider two main baselines: (i) *Random Baseline* does not use the gradient updates and simply randomly samples reconstructions from the per-feature marginals of the input dataset. Due to the structure of tabular datasets, we can easily estimate the marginal distribution of each feature. For the categorical features this can be done by simple counting, and for the continuous features we do it by defining a binning scheme with 100 equally spaced bins between the lower and upper bounds of the feature. Although this baseline is usually not realizable in practice (as it assumes prior knowledge of the marginals), it helps us calibrate our metric as performing below this baseline signals that there is no information being extracted from the client updates. Note that because both the selection of a batch and the random baseline represent sampling from the (approximate) data generating distribution, the random baseline monotonously increases in accuracy with growing batch size, (ii) *Cosine Baseline* is based on the work of Geiping et al. (2020), who established a strong attack for images. We transfer their attack to tabular data by removing the total variation prior used for images. Note that in the case of most competitive attacks on image and text, when removing the domain specific elements, they reduce to this baseline, therefore it is a reasonable choice for benchmarking a new domain.

**Experimental Setup**  For all attacks, we use the Adam optimizer (Kingma & Ba, 2014) with learning rate 0.06 for 1 500 iterations and without a learning rate schedule to perform the optimization in Alg. 1. In line with Geiping et al. (2020), we modify the update step of the optimizer by reducing the update gradient to its element-wise sign. The neural network we attack is a fully connected neural network with two hidden layers of 100 neurons each. We conducted our experiments on four popular mixed-type tabular binary classification datasets, the Adult census dataset (Dua & Graff, 2017), the German Credit dataset (Dua & Graff, 2017), the Lawschool Admission dataset (Wightman, 2017), and the Health Heritage dataset from Kaggle[2]. Due to the space constraints, here we report only our results on the Adult dataset, and refer the reader to App. D for full results on all four datasets. Finally, for all reported numbers below, we attack a neural network at initialization and estimate the mean and standard deviation of each reported metric on 50 different batches. For experiments with varying network sizes and attacks against provable defenses, please see App. C. For further details on the experimental setup of each experiment, we refer the reader to App. B

**General Results against FedSGD**  In Tab. 1 we present the results of our strong attack TabLeak against FedSGD training, together with two ablation experiments, each time removing either the pooling (no pooling) or the softmax component (no softmax). We compare our results to the baselines introduced above, on batch sizes 8, 16, 32, 64, and 128, once assuming knowledge of the true labels (top) and once using labels reconstructed by the method of Geng et al. (2021) (bottom). Notice that the noisy label reconstruction only influences the results for lower batch sizes, and manifests itself mostly in higher variance in the results. It is also worth to note that for batch size 8 (and lower, see App. D) all attacks can recover almost all the data, exposing a trivial vulnerability of FL on tabular

---

[2]Source: https://www.kaggle.com/c/hhp

Table 1: The mean inversion accuracy [%] and standard deviation of different methods over varying batch sizes with given true labels (True $y$) and with reconstructed labels (Rec. $\hat{y}$) on the Adult dataset.

| Label | Batch Size | TabLeak | TabLeak (no pooling) | TabLeak (no softmax) | Cosine | Random |
|---|---|---|---|---|---|---|
| True $y$ | 8 | **95.1 ± 9.2** | 93.9 ± 10.2 | 92.9 ± 6.5 | 91.1 ± 7.3 | 53.9 ± 4.4 |
| | 16 | **89.5 ± 7.6** | 84.5 ± 9.9 | 80.5 ± 4.3 | 75.0 ± 5.2 | 55.1 ± 3.9 |
| | 32 | **77.6 ± 4.8** | 72.4 ± 4.6 | 70.8 ± 3.2 | 66.6 ± 3.5 | 58.0 ± 2.9 |
| | 64 | **71.2 ± 2.8** | 66.2 ± 2.8 | 66.9 ± 2.7 | 62.5 ± 3.1 | 59.0 ± 3.2 |
| | 128 | **68.8 ± 1.3** | 64.1 ± 1.4 | 64.0 ± 2.1 | 59.5 ± 2.1 | 61.2 ± 3.1 |
| Rec. $\hat{y}$ | 8 | **86.9 ± 11.6** | 84.6 ± 13.4 | 85.8 ± 9.9 | 83.3 ± 9.7 | 53.9 ± 4.4 |
| | 16 | **82.4 ± 8.4** | 78.3 ± 9.0 | 77.7 ± 4.1 | 73.0 ± 3.5 | 55.1 ± 3.9 |
| | 32 | **75.3 ± 4.8** | 70.6 ± 4.3 | 70.2 ± 3.2 | 66.3 ± 3.4 | 58.0 ± 2.9 |
| | 64 | **70.4 ± 3.2** | 65.9 ± 3.6 | 66.8 ± 2.6 | 63.1 ± 3.2 | 59.0 ± 3.2 |
| | 128 | **68.7 ± 1.3** | 64.4 ± 1.5 | 63.8 ± 2.1 | 59.5 ± 2.1 | 61.2 ± 3.1 |

data. In case of larger batch sizes, even up to 128, TabLeak can recover a significant portion of the client's private data, well above random guessing, while the baseline Cosine attack fails to do so, demonstrating the necessity of a domain tailored attack. In a later paragraph, we show how we can further improve our reconstruction on this batch size and extract subsets of features with $> 90\%$ accuracy using the entropy. Further, the results on the ablation attacks demonstrate the effectiveness of each attack component, both providing a non-trivial improvement over the baseline attack that is preserved when combined in our strongest attack. Demonstrating generalization beyond Adult, we include our results on the German Credit, Lawschool Admissions, and Health Heritage datasets in App. D.1, where we also outperform the Cosine baseline attack by at least $10\%$ on batch size 32 on each dataset.

**Categorical vs. Continuous Features**    An interesting effect of having mixed type features in the data is that the reconstruction success clearly differs by feature type. As we can observe in Fig. 3, the continuous features produce an up to $30\%$ lower accuracy than the categorical features for the same batch size. We suggest that this is due to the nature of categorical features and how they are encoded. While trying to match the gradients by optimizing the reconstruction, having the correct categorical features will have a much greater effect on the gradient alignment, as when encoded, they take up the majority of the data vector. Also, when reconstructing a one-hot encoded categorical feature, we only have to be able to retrieve the location of the maximum in a vector of length $D_i$, whereas for the successful reconstruction of a continuous feature we have to retrieve its value correctly up to a small error. Therefore, especially when the

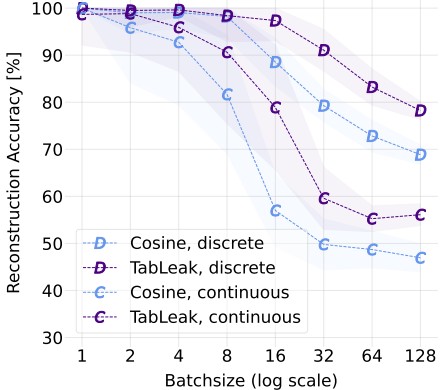

Figure 3: The inversion accuracy on the Adult dataset over varying batch size separated for discrete (D) and continuous (C) features.

optimization process is aware of the structure of the encoding scheme (*e.g.,* by using the softmax structural prior), categorical features are much easier to reconstruct. This poses a critical privacy risk in tabular federated learning, as sensitive features are often categorical, *e.g.,* gender or race.

**Federated Averaging**    In training with FedAvg (McMahan et al., 2016) participating clients conduct local training of several updates before communicating their new parameters to the server. Note that the more local updates are conducted by the clients, the harder a reconstruction attack becomes, making leakage attacks against FedAvg more challenging. Although this training method is of significantly higher practical importance than FedSGD, most prior work does not evaluate against it. Building upon the work of Dimitrov et al. (2022b) (for details please see App. B and the work of Dimitrov et al. (2022b)), we evaluate our combined attack and the cosine baseline in the setting of Federated Averaging. We present our results of retrieving a client dataset of size 32 over varying number of local batches and epochs on the Adult dataset in Tab. 2, while assuming full knowledge of the true labels. We observe that while our combined attack significantly outperforms the random

Table 2: Mean and standard deviation of the inversion accuracy [%] on FedAvg with local dataset sizes of 32 on the Adult dataset. The accuracy of the random baseline for 32 datapoints is $58.0 \pm 2.9$.

| | TabLeak | | | Cosine | | |
|---|---|---|---|---|---|---|
| n. batches | 1 epoch | 5 epochs | 10 epochs | 1 epoch | 5 epochs | 10 epochs |
| 1 | $\mathbf{77.4 \pm 4.5}$ | $\mathbf{71.1 \pm 2.9}$ | $\mathbf{67.6 \pm 3.7}$ | $65.2 \pm 2.7$ | $56.1 \pm 4.1$ | $53.2 \pm 4.2$ |
| 2 | $\mathbf{75.7 \pm 5.0}$ | $\mathbf{71.7 \pm 3.9}$ | $\mathbf{67.7 \pm 4.2}$ | $64.8 \pm 3.3$ | $56.4 \pm 4.8$ | $56.2 \pm 4.8$ |
| 4 | $\mathbf{75.9 \pm 4.4}$ | $\mathbf{71.0 \pm 3.2}$ | $\mathbf{67.4 \pm 3.4}$ | $64.8 \pm 3.4$ | $58.7 \pm 4.6$ | $56.6 \pm 5.0$ |

Table 3: The mean accuracy [%] and entropies with the corresponding standard deviations over batch sizes of the categorical (top) and continuous (bottom) features.

| | Batch Size | | | | |
|---|---|---|---|---|---|
| | 8 | 16 | 32 | 64 | 128 |
| Acc. | $98.5 \pm 5.6$ | $97.2 \pm 4.3$ | $91.0 \pm 4.4$ | $83.2 \pm 3.6$ | $78.5 \pm 1.8$ |
| $\bar{H}^D$ | $0.15 \pm 0.13$ | $0.26 \pm 0.11$ | $0.40 \pm 0.06$ | $0.48 \pm 0.04$ | $0.53 \pm 0.03$ |
| Acc. | $90.9 \pm 14.7$ | $78.8 \pm 13.5$ | $59.2 \pm 6.9$ | $55.1 \pm 3.0$ | $55.7 \pm 2.0$ |
| $H^C$ | $-1.11 \pm 0.95$ | $-0.11 \pm 0.63$ | $0.77 \pm 0.30$ | $1.21 \pm 0.19$ | $1.48 \pm 0.10$ |

baseline of $58.0\%$ accuracy even up to 40 local updates, the baseline attack fails to consistently do so whenever the local training is longer than one epoch. As FedAvg with tabular data is of high practical relevance, our results which highlight its vulnerability are concerning. We show further details for the experimental setup and results on other datasets in App. B and App. D, respectively.

**Assessing Reconstructions via Entropy** We now investigate how an adversary can use the entropy (introduced in Sec. 3.3) to assess the quality of their reconstructions. In Tab. 3 we show the mean and standard deviation of the accuracy and the entropy of both the discrete and the continuous features over increasing batch sizes after reconstructing with TabLeak (ensemble size 30). We observe an increase in the mean entropy over the increasing batch sizes, corresponding to accuracy decrease in the reconstructed batches. Hence, an attacker can understand the global effectiveness of their attack by looking at the retrieved entropies, without having to compare their results to the ground truth.

Table 4: The mean accuracy [%] and the share of data [%] in each entropy bucket for batch size 128 on the Adult dataset.

| Entropy | Categorical Features | |
|---|---|---|
| Bucket | Accuracy [%] | Data [%] |
| 0.0-0.2 | 95.7 | 8.1 |
| 0.2-0.4 | 90.5 | 23.4 |
| 0.4-0.6 | 79.8 | 27.7 |
| 0.6-0.8 | 69.8 | 29.2 |
| 0.8-1.0 | 61.2 | 11.6 |
| Overall | 78.5 | 100 |
| Random | 73.8 | 100 |

We now look at a single batch of size 128 and put each categorical feature into a bucket based on their reconstruction entropy after attacking with TabLeak (ensemble size 30). In Tab. 4 we present our results, showing that features falling into lower entropy buckets (0.0-0.2 and 0.2-0.4) inside a batch are significantly more accurately reconstructed ($> 90\%$) than the overall batch ($78.5\%$). Note that this bucketing can be done without the knowledge of the ground-truth, yet the adversary can concretely identify the high-fidelity features in their noisy reconstruction. This shows that even for reconstructions of large batches that seem to contain little-to-no information (close to random baseline), an adversary can still extract subsets of the data with high accuracy. Tables containing both feature types on all four datasets can be found in App. D.4, providing analogous conclusions.

## 5 CONCLUSION

In this work we presented TabLeak, the first data leakage attack on tabular data in the setting of federated learning (FL), obtaining state-of-the-art results against both popular FL training protocols in the tabular domain. As tabular data is ubiquitous in privacy critical high-stakes applications, our results raise important concerns regarding practical systems currently using FL. Therefore, we advocate for further research on advancing defenses necessary to mitigate such privacy leaks.

## 6 Ethics Statement

As tabular data is often used in high-stakes applications and may contain sensitive data of natural or legal persons, confidential treatment is critical. This work presents an attack algorithm in the tabular data domain that enables an FL server to steal the private data of its clients in industry-relevant scenarios, deeming such applications potentially unsafe.

We believe that exposing vulnerabilities of both recently proposed and widely adopted systems, where privacy is a concern, can benefit the development of adequate safety mechanisms against malicious actors. In particular, this view is shared by the governmental institutions of the United States of America and the United Kingdom that jointly supported the launch of a competition (`https://petsprizechallenges.com/`) aimed at advancing the privacy of FL in the tabular domain, encouraging the participation of both teams developing defenses and attacks. Also, as our experiments in App. C.1 show, existing techniques can help mitigate the privacy threat, hence we encourage practitioners to make use of them.

## 7 Reproducibility Statement

We publish all of our code and provide it to the reviewers in an anonymized form on the openreview discussion forum. Alongside the code, we provide a detailed README file with instructions on how to run all of our experiments presented in Sec. 4 and our additional experiments presented in App. C. To allow better reproducibility, our code presets random seeds to the chosen constant $42$. Further, all of our reported numbers are statistics of $50$ independent repeated experiments, and wherever possible, we report standard deviations. Finally, our hyperparameters and information about the hardware on which we ran our experiments are provided in App. B.

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

Table 6: Dataset specifications.

|  | Features | Discrete Features | Continuous Features | Encoded Features | Data Points |
|---|---|---|---|---|---|
| Adult | 14 | 8 | 6 | 105 | 45 222 |
| German | 20 | 13 | 7 | 63 | 1 000 |
| Lawschool | 7 | 5 | 2 | 39 | 96 584 |
| Health Heritage | 17 | 6 | 11 | 110 | 218 415 |

## A  ACCURACY METRIC

To ease the understanding, we start by repeating our accuracy metric here, where we measure the reconstruction accuracy between the retrieved sample $\hat{x}$ and the ground truth $x$ as:

$$\text{accuracy}(x, \hat{x}) := \frac{1}{K + L} \left( \sum_{i=1}^{K} \mathbb{I}\{x_i^D = \hat{x}_i^D\} + \sum_{i=1}^{L} \mathbb{I}\{\hat{x}_i^C \in [x_i^C - \epsilon_i, \, x_i^C + \epsilon_i]\} \right). \quad (8)$$

Note that the binary treatment of continuous features in our accuracy metric enables the combined measurement of the accuracy on both the discrete and the continuous features. From an intuitive point of view, this measure closely resembles how one would judge the correctness of numerical guesses. For example, guessing the age of a 25 year old, one would deem the guess good if it is within 3 to 4 years of the true value, but the guesses 65 and 87 would be both qualitatively incorrect. In order to facilitate scalability of our experiments, we chose the $\{\epsilon_i\}_{i=1}^{L}$ error-tolerance-bounds based on the global standard deviation if the given continuous feature $\sigma_i^C$ and multiplied it by a constant, concretely, we used $\epsilon_i = 0.319 \, \sigma_i^C$ for all our experiments. Note that $Pr[\mu - 0.319 \, \sigma < x < \mu + 0.319 \, \sigma] \approx 0.25$ for a Gaussian random variable $x$ with mean $\mu$ and variance $\sigma^2$. For our metric this means that assuming Gaussian zero-mean error in the reconstruction around the true value, we accept our reconstruction as privacy leakage as long as we fall into the 25% error-probability range around the correct value. In Tab. 5 we list the tolerance bounds $\epsilon_i$ for the continuous features of the Adult dataset produced by this method. We would like to remark here, that we fixed our metric parameters before conducting any experiments, and did not adjust them based on any obtained results. Note also that in App. C we provide results where the continuous feature reconstruction accuracy is measured using the commonly used regression metric of root mean squared error (RMSE), where TabLeak also achieves the best results, signaling that the success of our method is independent of our chosen metric.

Table 5: Resulting tolerance bounds on the Adult dataset when using $\epsilon_i = 0.319 \, \sigma_i^C$, as used by us for our experiments.

| feature | age | fnlwgt | education-num | capital-gain | capital-loss | hours-per-week |
|---|---|---|---|---|---|---|
| tolerance | 4.2 | 33699 | 0.8 | 2395 | 129 | 3.8 |

## B  FURTHER EXPERIMENTAL DETAILS

Here we give an extended description to our experimental details provided in Sec. 4, additionally we provide the specifications of each used dataset in Tab. 6. For all attacks, we use the Adam optimizer (Kingma & Ba, 2014) with learning rate 0.06 for 1 500 iterations and without a learning rate schedule. We chose the learning rate based on our experiments on the baseline attack where it performed best. In line with Geiping et al. (2020), we modify the update step of the optimizer by reducing the update gradient to its element-wise sign. We attack a fully connected neural network with two hidden layers of 100 neurons each at initialization. However, we provide a network-size ablation in Fig. 8, where we evaluate our attack against the baseline method for 5 different network architectures. For each reported metric we conduct 50 independent runs on 50 different batches to estimate their statistics. For all FedSGD experiments we clamp the continuous features to their valid ranges before measuring the reconstruction accuracy, both for our attacks and the baseline methods. We ran each of our experiments on single cores of Intel(R) Xeon(R) CPU E5-2690 v4 @ 2.60GHz.

**Federated Averaging Experiments** For experiments on attacking the FedAvg training algorithm, we fix the clients' local dataset size at 32 and conduct an attack after local training with learning rate 0.01 on the initialized network described above. We use the FedAvg attack-framework of Dimitrov et al. (2022b), where for each local training epoch we initialize an independent mini-dataset matching the size of the client dataset, and simulate the local training of the client. At each reconstruction update, we use the mean squared error between the different epoch data means ($D_{\text{inv}} = \ell_2$ and $g = $ mean in Dimitrov et al. (2022b)) as the permutation invariant epoch prior required by the framework, ensuring the consistency of the reconstructed dataset. For the full technical details, please refer to the manuscript of Dimitrov et al. (2022b). For choosing the prior parameter $\lambda_{\text{inv}}$, we conduct line-search on each setup and attack method pair individually on the parameters $[0.0, 0.5, 0.1, 0.05, 0.01, 0.005, 0.001]$, and pick the ones providing the best results. Further, to reduce computational overhead, we reduce the ensemble size of TabLeak from 30 to 15 for these experiments on all datasets.

## C  FURTHER EXPERIMENTS

In this subsection, we present three further experiments:

- Results of attacking neural networks defended using differentially private noisy gradients in App. C.1.

- Ablation study on the impacts of the neural network's size on the reconstruction difficulty in App. C.2.

- Measuring the Root Mean Squared Error (RMSE) of the reconstruction of continuous features in App. C.3.

### C.1  ATTACK AGAINST GAUSSIAN DP

Differential privacy (DP) has recently gained popularity, as a way to prevent privacy violations in FL (Abadi et al., 2016; Zhu et al., 2019). Unlike, empirical defenses which are often broken by specifically crafted adversaries (Balunović et al., 2021), DP provides guarantees on the amount of data leaked by a FL model, in terms of the magnitude of random noise the clients add to their gradients prior to sharing them with the server (Abadi et al., 2016; Zhu et al., 2019). Naturally, DP methods balance privacy concerns with the accuracy of the produced model, since bigger noise results in worse models that are more private. In this subsection, we evaluate TabLeak, and the Cosine baseline against DP defended gradient updates, where zero-mean Gaussian noise is added with standard deviations 0.001, 0.01, and 0.1 to the client gradients. We present our results on the Adult, German Credit, Lawschool Admissions, and Health Heritage datasets in Fig. 4, Fig. 5, Fig. 6, and Fig. 7, respectively. Although both methods are affected by the defense, our method consistently produces better reconstructions than the baseline method. However, for high noise level (standard deviation = 0.1) and larger batch size both attacks break, advocating for the use of DP defenses in tabular FL to prevent the high vulnerability exposed by this work.

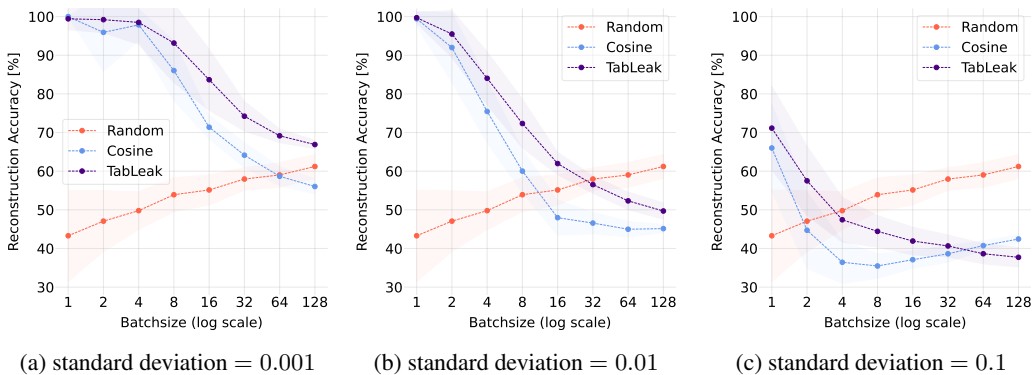

Figure 4: Mean and standard deviation accuracy [%] curves over batch size at varying Gaussian noise level $\sigma$ added to the client gradients for differential privacy on the **Adult** dataset.

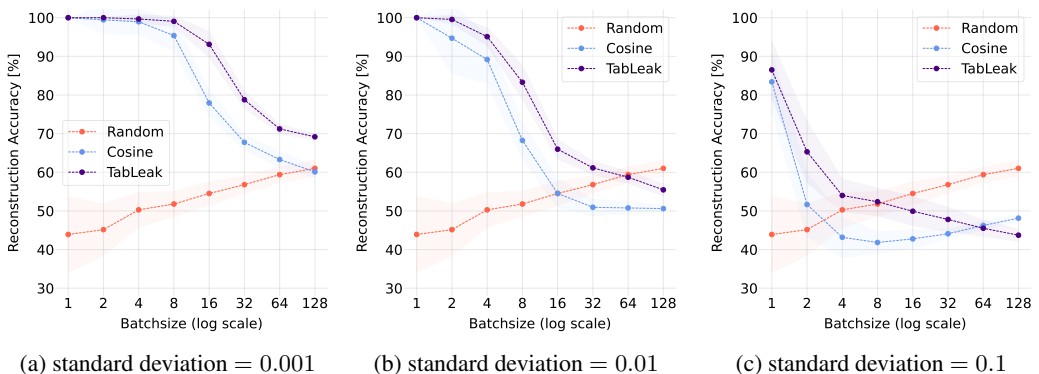

Figure 5: Mean and standard deviation accuracy [%] curves over batch size at varying Gaussian noise level $\sigma$ added to the client gradients for differential privacy on the **German Credit** dataset.

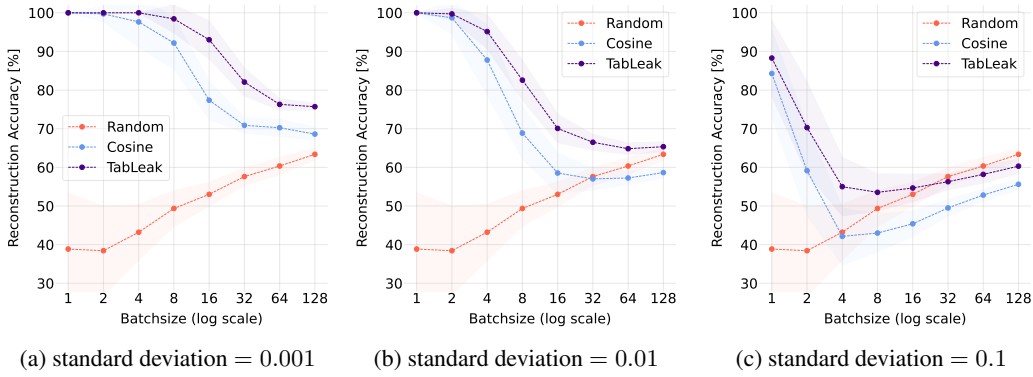

Figure 6: Mean and standard deviation accuracy [%] curves over batch size at varying Gaussian noise level $\sigma$ added to the client gradients for differential privacy on the **Lawschool Admissions** dataset.

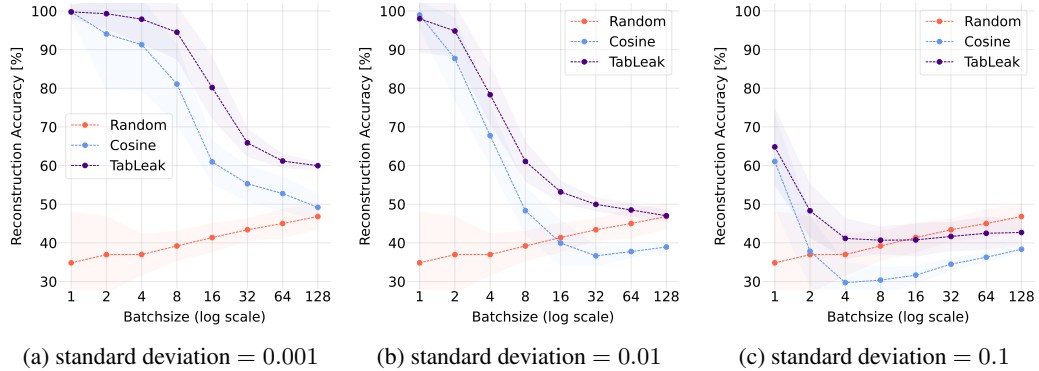

(a) standard deviation $= 0.001$     (b) standard deviation $= 0.01$     (c) standard deviation $= 0.1$

Figure 7: Mean and standard deviation accuracy [%] curves over batch size at varying Gaussian noise level $\sigma$ added to the client gradients for differential privacy on the **Health Heritage** dataset.

## C.2 VARYING NETWORK SIZE

To understand the effect the choice of the network has on the obtained reconstruction results, we defined 4 additional fully connected networks, two smaller, and two bigger ones to evaluate TabLeak on. As a simple linear model is often a good baseline for tabular data, we add it also to the range of attacked models. Concretely, we examined the following six models for our attack:

- Linear: a linear classification network: $f_W(c(x)) = \sigma(Wc(x) + b)$,
- NN 1: a single hidden layer neural network with 50 neurons,
- NN 2: a single hidden layer neural network with 100 neurons,
- NN 3: a neural network with two hidden layers of 100 neurons each (network used in main body),
- NN 4: a neural network with three hidden layers of 200 neurons each,
- NN 5: a three hidden layer neural network with 400 neurons in each layer.

We attack the above networks, aiming to reconstruct a batch of size 32. We plot the accuracy of TabLeak and the cosine baseline as a function of the number of parameters in the network in Fig. 8 for all four datasets. We can observe that with increasing number of parameters in the network, the reconstruction accuracy significantly increases on all datasets, and rather surprisingly, allowing for near perfect reconstruction of a batch as large as 32 in some cases. Observe that on both ends of the presented parameter scale the differences between the methods degrade, *i.e.* they either both converge to near-perfect reconstruction (large networks) or to random guessing (small networks). Therefore, the choice of our network for conducting the experiments was instructive in examining the differences between the methods.

Additionally, to better understand the relevance of the models examined here, we train them on each of the datasets for 50 epochs and observe their behavior through monitoring their performance on a secluded test set of each dataset. We do this for 5 different initializations of each model, and report the mean and the standard deviation of the test accuracy at each training epoch for each model. Note that we do not train the models using any FL protocol, merely, this experiment serves to give a better understanding between the relation of the given dataset and the model used, putting also the attack success data in better perspective. For training, we use the Adam Kingma & Ba (2014) optimizer and batch size 256 for each of the datasets, except for the German Credit dataset, where we train with batch size 64 due to its small size. We provide all test accuracy curves over training in Fig. 9. From the accuracy curves we can observe that most large models that are easy to attack tend to overfit quickly to the data, indicating a heavily overparameterized regime. Additionally, in Tab. 7 we provide the peak mean test accuracies per dataset and model, effectively corresponding to a 'perfect' early-stopping. The linear model could appear to be an overall good choice, as it is very hard to attack and shows good stability during training, however, it does not achieve competitive performance on most datasets. In Tab. 7 the non-linear models always outperform the linear model, and achieve comparable performance across themselves in this ideal setting, where overfitting can be prevented

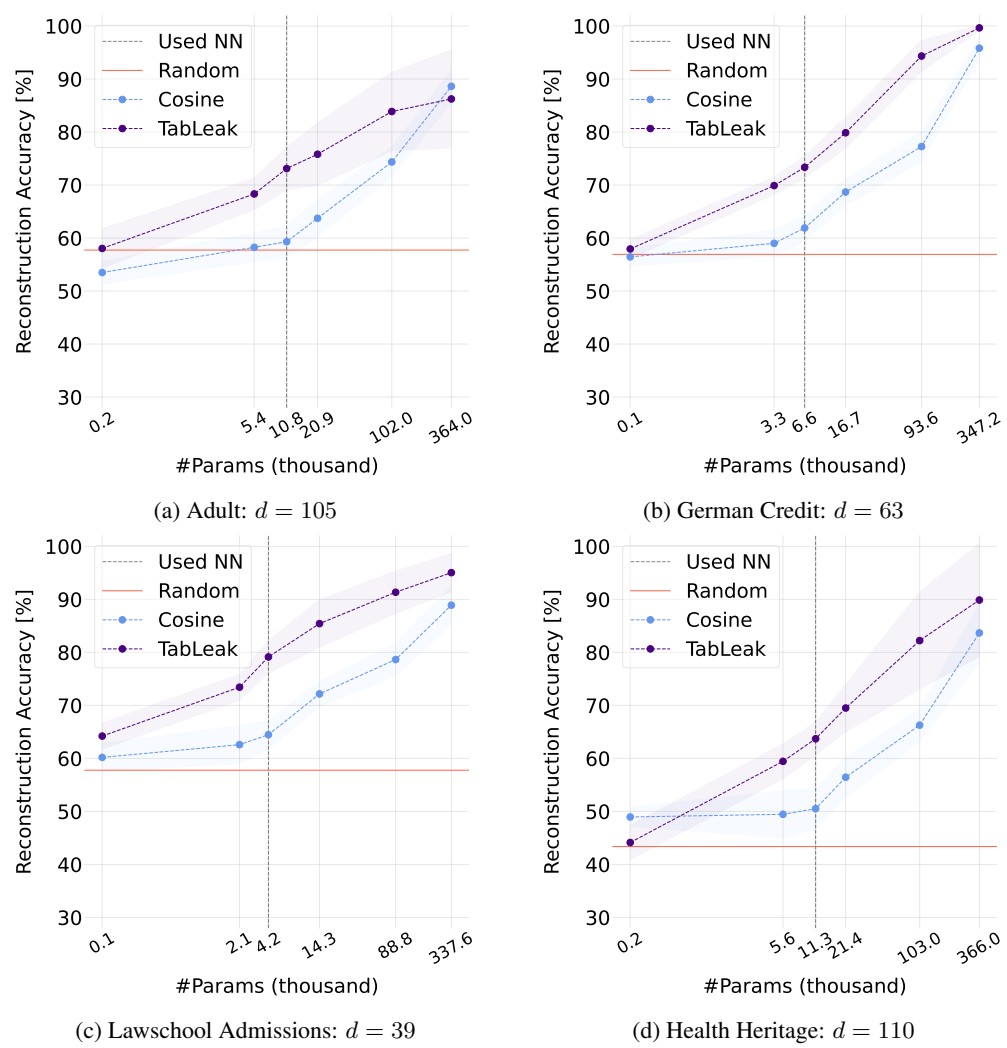

(a) Adult: $d = 105$

(b) German Credit: $d = 63$

(c) Lawschool Admissions: $d = 39$

(d) Health Heritage: $d = 110$

Figure 8: Mean attack accuracy curves with standard deviation for batch size 32 over varying network size (measured in number of parameters, #Params, log scale) on all four datasets with $d$ number of features after encoding. We mark the network we used for our other experiments with a dashed vertical line. From left to right we have the following models: Linear, NN 1, NN 2, NN 3, NN 4, and NN5.

Table 7: Mean and standard deviation of the peak test accuracy of each of the examined 6 models on the four discussed datasets over training.

|                 | Linear       | Layout 1         | Layout 2         | Layout 3         | Layout 4         | Layout 5     |
|-----------------|--------------|------------------|------------------|------------------|------------------|--------------|
| Adult           | $84.7 \pm 0.1$ | $84.9 \pm 0.1$   | $\mathbf{85.0 \pm 0.1}$ | $84.8 \pm 0.1$   | $84.8 \pm 0.1$   | $84.7 \pm 0.1$ |
| German          | $73.0 \pm 1.4$ | $80.0 \pm 1.1$   | $79.5 \pm 0.6$   | $\mathbf{80.9 \pm 0.7}$ | $78.9 \pm 1.0$   | $79.4 \pm 1.8$ |
| Lawschool       | $87.4 \pm 0.0$ | $89.6 \pm 0.1$   | $89.8 \pm 0.0$   | $\mathbf{90.0 \pm 0.1}$ | $89.8 \pm 0.1$   | $89.8 \pm 0.1$ |
| Health Heritage | $80.9 \pm 0.0$ | $\mathbf{81.2 \pm 0.1}$ | $\mathbf{81.2 \pm 0.0}$ | $81.2 \pm 0.1$   | $81.2 \pm 0.1$   | $81.1 \pm 0.1$ |

by monitoring on the test data[3]. Conclusively, simpler non-linear models shall be pursued for FL on tabular data, as they are less prone to overfitting and provide better protection from data leakage attacks.

---

[3] In practice a proxy metric would be necessary to achieve early-stopping, such as monitoring the performance on a separate validation set split from the training data

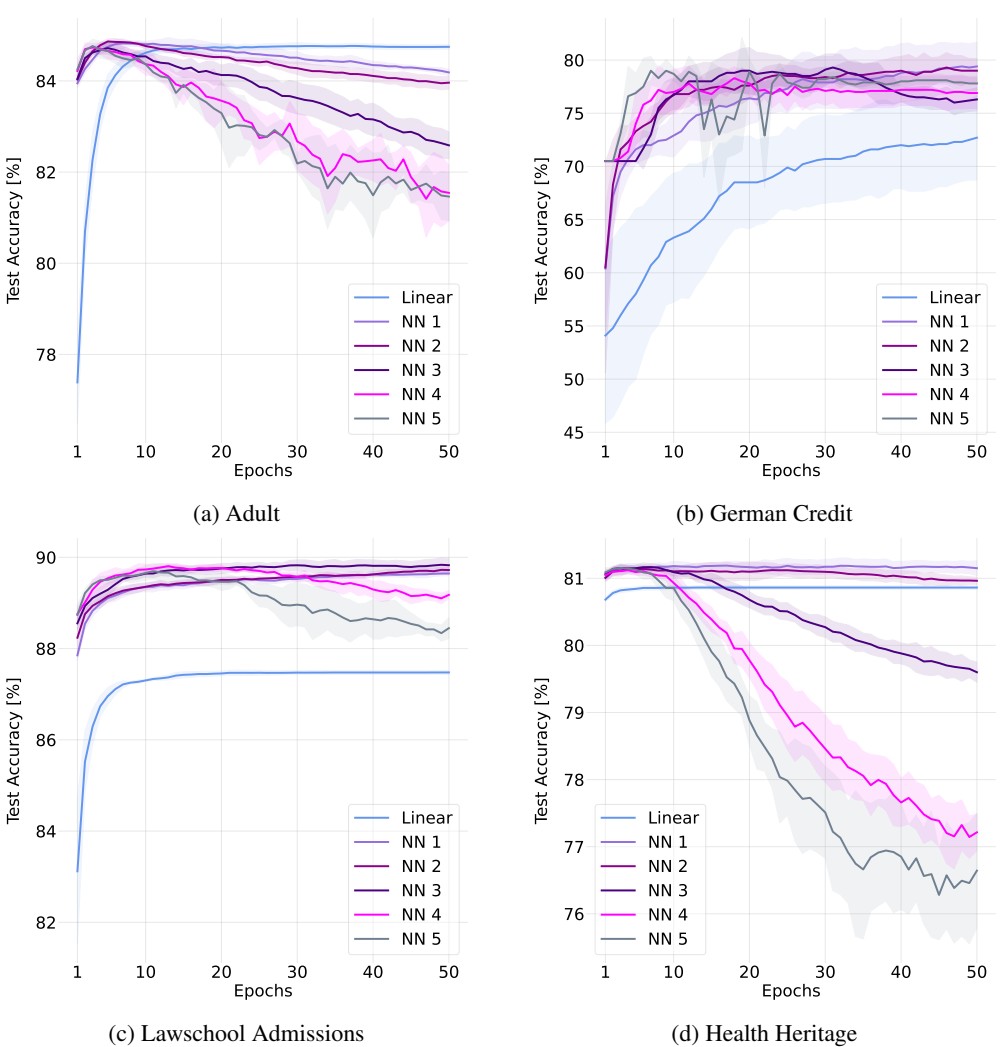

(a) Adult

(b) German Credit

(c) Lawschool Admissions

(d) Health Heritage

Figure 9: Mean test and standard deviation of the test accuracy over epochs during five independent runs of training for each examined model on all four datasets. For our experiments elsewhere we used the network corresponding to Layout 3, marked in red here.

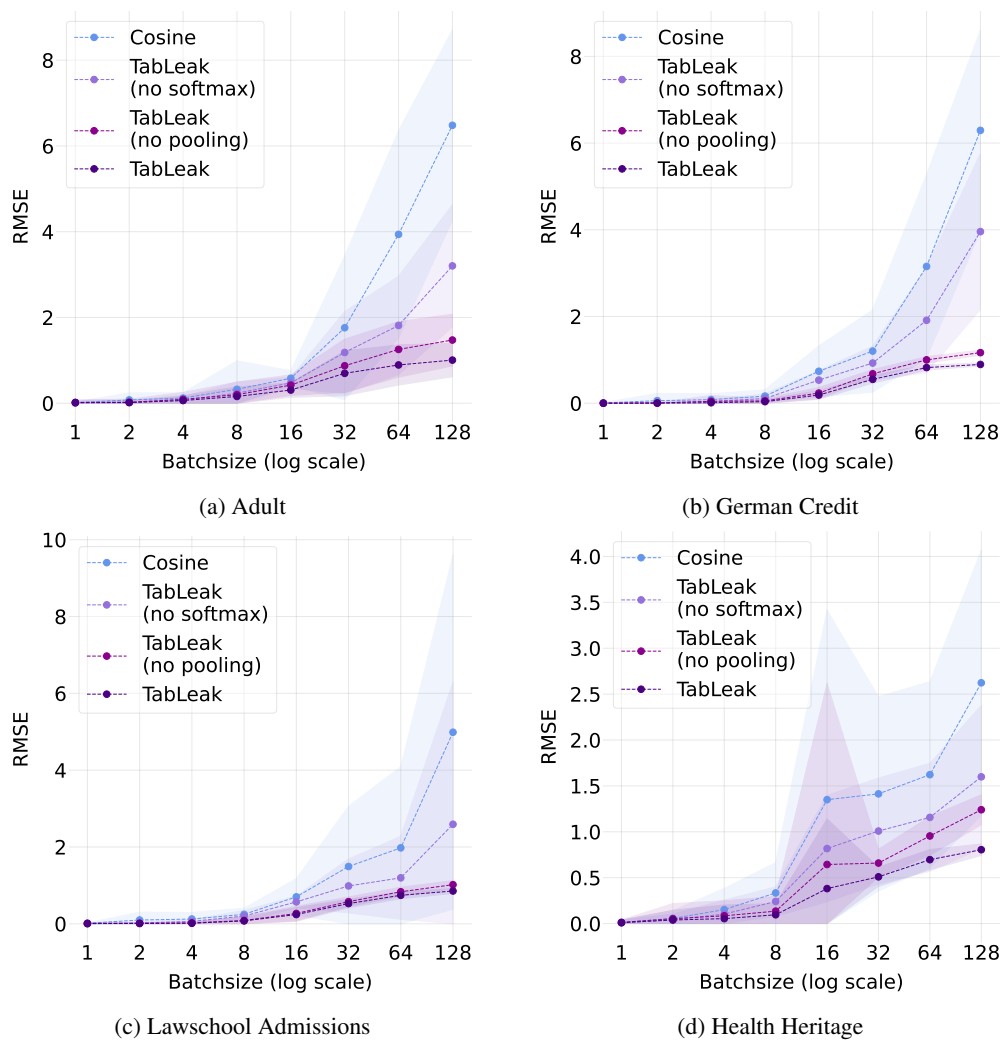

Figure 10: The mean and standard deviation of the Root Mean Square Error (RMSE) of the reconstructions of the continuous features on all four datasets over batch sizes.

## C.3 Continuous Feature Reconstruction Measured by RMSE

In order to examine the potential influence of our choice of reconstruction metric on the obtained results, we further measured the reconstruction quality of continuous features on the widely used Root Mean Squared Error (RMSE) metric as well. Concretely, we calculate the RMSE between the $L$ continuous features of our reconstruction $\hat{x}^C$ and the ground truth $x$ in a batch of size $n$ as:

$$\text{RMSE}(x^C, \hat{x}^C) = \frac{1}{L} \sum_{i=1}^{L} \sqrt{\frac{1}{n} \sum_{j=1}^{n} (x_{ij}^C - \hat{x}_{ij}^C)^2}. \quad (9)$$

As our results in Fig. 10 demonstrate, TabLeak achieves significantly lower RMSE than the Cosine baseline on large batch sizes, for all four datasets examined. This indicates that the strong results obtained by TabLeak in the rest of the paper are not a consequence of our evaluation metric.

Table 8: The mean inversion accuracy [%] and standard deviation of different methods over varying batch sizes with given true labels (top) and with reconstructed labels (bottom) on the **Adult** dataset.

| Label | Batch Size | TabLeak | TabLeak (no pooling) | TabLeak (no softmax) | Cosine | Random |
|---|---|---|---|---|---|---|
| True $y$ | 1 | $99.4 \pm 2.8$ | $99.1 \pm 4.4$ | $\mathbf{100.0 \pm 0.0}$ | $\mathbf{100.0 \pm 0.0}$ | $43.3 \pm 11.8$ |
| | 2 | $99.2 \pm 5.5$ | $99.1 \pm 6.5$ | $\mathbf{99.9 \pm 1.0}$ | $97.6 \pm 6.9$ | $47.1 \pm 7.9$ |
| | 4 | $98.0 \pm 4.5$ | $96.6 \pm 7.5$ | $\mathbf{98.9 \pm 4.0}$ | $96.4 \pm 7.2$ | $49.8 \pm 4.9$ |
| | 8 | $\mathbf{95.1 \pm 9.2}$ | $93.9 \pm 10.2$ | $92.9 \pm 6.5$ | $91.1 \pm 7.3$ | $53.9 \pm 4.4$ |
| | 16 | $\mathbf{89.5 \pm 7.6}$ | $84.5 \pm 9.9$ | $80.5 \pm 4.3$ | $75.0 \pm 5.2$ | $55.1 \pm 3.9$ |
| | 32 | $\mathbf{77.6 \pm 4.8}$ | $72.4 \pm 4.6$ | $70.8 \pm 3.2$ | $66.6 \pm 3.5$ | $58.0 \pm 2.9$ |
| | 64 | $\mathbf{71.2 \pm 2.8}$ | $66.2 \pm 2.8$ | $66.9 \pm 2.7$ | $62.5 \pm 3.1$ | $59.0 \pm 3.2$ |
| | 128 | $\mathbf{68.8 \pm 1.3}$ | $64.1 \pm 1.4$ | $64.0 \pm 2.1$ | $59.5 \pm 2.1$ | $61.2 \pm 3.1$ |
| Rec. $\hat{y}$ | 1 | $99.4 \pm 2.8$ | $99.3 \pm 3.6$ | $\mathbf{100.0 \pm 0.0}$ | $\mathbf{100.0 \pm 0.0}$ | $43.3 \pm 11.8$ |
| | 2 | $98.2 \pm 8.9$ | $98.1 \pm 9.1$ | $\mathbf{98.6 \pm 7.5}$ | $95.9 \pm 11.5$ | $47.1 \pm 7.9$ |
| | 4 | $89.5 \pm 13.8$ | $88.0 \pm 15.2$ | $\mathbf{90.0 \pm 13.0}$ | $87.9 \pm 13.7$ | $49.8 \pm 4.9$ |
| | 8 | $\mathbf{86.9 \pm 11.6}$ | $84.6 \pm 13.4$ | $85.8 \pm 9.9$ | $83.3 \pm 9.7$ | $53.9 \pm 4.4$ |
| | 16 | $\mathbf{82.4 \pm 8.4}$ | $78.3 \pm 9.0$ | $77.7 \pm 4.1$ | $73.0 \pm 3.5$ | $55.1 \pm 3.9$ |
| | 32 | $\mathbf{75.3 \pm 4.8}$ | $70.6 \pm 4.3$ | $70.2 \pm 3.2$ | $66.3 \pm 3.4$ | $58.0 \pm 2.9$ |
| | 64 | $\mathbf{70.4 \pm 3.2}$ | $65.9 \pm 3.6$ | $66.8 \pm 2.6$ | $63.1 \pm 3.2$ | $59.0 \pm 3.2$ |
| | 128 | $\mathbf{68.7 \pm 1.3}$ | $64.4 \pm 1.5$ | $63.8 \pm 2.1$ | $59.5 \pm 2.1$ | $61.2 \pm 3.1$ |

# D  ALL MAIN RESULTS

In this subsection, we include all the results presented in the main part of this paper for the Adult dataset alongside with the corresponding additional results on the German Credit, Lawschool Admissions, and the Health Heritage datasets.

## D.1  FULL FEDSGD RESULTS ON ALL DATASETS

In Tab. 8, Tab. 9, Tab. 10, and Tab. 11 we provide the full attack results of our method compared to the Cosine and random baselines on the Adult, German Credit, Lawschool Admissions, and Health Heritage datasets, respectively. Looking at the results for all datasets, we can confirm the observations made in Sec. 4, *i.e.* (i) the lower batch sizes are vulnerable to any non-trivial attack, (ii) not knowing the ground truth labels does not significantly disadvantage the attacker for larger batch sizes, and (iii) TabLeak provides a strong improvement over the baselines for practically relevant batch sizes over all datasets examined.

## D.2  CATEGORICAL VS. CONTINUOUS FEATURES ON ALL DATASETS

In Fig. 11, we compare the reconstruction accuracy of the continuous and the discrete features on all four datasets. We confirm our observations, shown in Fig. 3 in the main text, that a strong dichotomy between continuous and discrete feature reconstruction accuracy exists on all 4 datasets.

## D.3  FEDERATED AVERAGING RESULTS ON ALL DATASETS

In Tab. 12, Tab. 13, Tab. 14, and Tab. 15 we present our results on attacking the clients in FedAvg training on the Adult, German Credit, Lawschool Submissions, and Health Heritage datasets, respectively. We described the details of the experiment in App. B above. Confirming our conclusions drawn in the main part of this manuscript, we observe that TabLeak achieves non-trivial reconstruction accuracy over all settings and even for large numbers of updates, while the baseline attack often fails to outperform random guessing, when the number of local updates is increased.

Table 9: The mean inversion accuracy [%] and standard deviation of different methods over varying batch sizes with given true labels (top) and with reconstructed labels (bottom) on the **German Credit** dataset.

| Label | Batch Size | TabLeak | TabLeak (no pooling) | TabLeak (no softmax) | Cosine | Random |
|---|---|---|---|---|---|---|
| True $y$ | 1 | **100.0 ± 0.0** | **100.0 ± 0.0** | **100.0 ± 0.0** | **100.0 ± 0.0** | 43.9 ± 9.8 |
| | 2 | **100.0 ± 0.0** | **100.0 ± 0.0** | 99.9 ± 0.7 | 98.0 ± 7.1 | 45.1 ± 6.6 |
| | 4 | **99.9 ± 0.4** | 99.2 ± 3.6 | 99.5 ± 1.2 | 97.8 ± 6.0 | 50.3 ± 4.5 |
| | 8 | 99.7 ± 1.1 | 99.1 ± 2.2 | **98.2 ± 2.5** | 96.1 ± 5.2 | 51.8 ± 3.2 |
| | 16 | **95.9 ± 3.4** | 94.0 ± 4.3 | 84.1 ± 3.4 | 79.3 ± 4.4 | 54.5 ± 3.0 |
| | 32 | **83.6 ± 2.9** | 79.4 ± 3.1 | 72.1 ± 1.9 | 69.7 ± 2.2 | 56.8 ± 2.2 |
| | 64 | **73.0 ± 1.3** | 70.8 ± 1.4 | 68.9 ± 1.4 | 66.6 ± 1.8 | 59.4 ± 1.9 |
| | 128 | **71.3 ± 0.8** | 69.1 ± 0.8 | 67.4 ± 1.5 | 64.5 ± 1.5 | 61.0 ± 2.1 |
| Rec. $\hat{y}$ | 1 | **100.0 ± 0.0** | **100.0 ± 0.0** | **100.0 ± 0.0** | **100.0 ± 0.0** | 43.9 ± 9.8 |
| | 2 | **100.0 ± 0.0** | 99.5 ± 3.5 | 99.9 ± 0.7 | 98.8 ± 5.2 | 45.1 ± 6.6 |
| | 4 | **99.6 ± 2.6** | 99.5 ± 2.9 | 99.2 ± 3.0 | 97.4 ± 6.4 | 50.3 ± 4.5 |
| | 8 | **97.2 ± 6.1** | 96.8 ± 6.8 | 96.0 ± 6.2 | 94.8 ± 6.5 | 51.8 ± 3.2 |
| | 16 | **91.7 ± 6.5** | 90.0 ± 7.3 | 82.3 ± 4.6 | 77.9 ± 4.6 | 54.5 ± 3.0 |
| | 32 | **81.5 ± 3.4** | 77.6 ± 2.8 | 71.5 ± 2.0 | 69.1 ± 2.1 | 56.8 ± 2.2 |
| | 64 | **72.9 ± 1.4** | 70.5 ± 1.4 | 68.6 ± 1.3 | 66.5 ± 1.7 | 59.4 ± 1.9 |
| | 128 | **71.1 ± 0.9** | 69.1 ± 0.7 | 67.1 ± 1.6 | 64.4 ± 1.6 | 61.0 ± 2.1 |

Table 10: The mean inversion accuracy [%] and standard deviation of different methods over varying batch sizes with given true labels (top) and with reconstructed labels (bottom) on the **Lawschool Admissions** dataset.

| Label | Batch Size | TabLeak | TabLeak (no pooling) | TabLeak (no softmax) | Cosine | Random |
|---|---|---|---|---|---|---|
| True $y$ | 1 | **100.0 ± 0.0** | **100.0 ± 0.0** | **100.0 ± 0.0** | **100.0 ± 0.0** | 38.9 ± 14.6 |
| | 2 | **100.0 ± 0.0** | **100.0 ± 0.0** | 99.9 ± 1.0 | 96.3 ± 10.4 | 38.4 ± 11.5 |
| | 4 | **100.0 ± 0.0** | **100.0 ± 0.0** | 99.7 ± 1.2 | 97.6 ± 6.9 | 43.2 ± 7.2 |
| | 8 | 98.7 ± 3.8 | **98.8 ± 3.7** | 96.0 ± 5.0 | 94.5 ± 5.8 | 49.4 ± 4.6 |
| | 16 | **94.8 ± 5.6** | 93.5 ± 6.5 | 81.1 ± 4.5 | 77.3 ± 5.5 | 53.0 ± 3.1 |
| | 32 | **84.8 ± 3.9** | 82.4 ± 4.1 | 73.3 ± 2.8 | 71.0 ± 2.8 | 57.6 ± 2.3 |
| | 64 | **78.2 ± 2.0** | 76.6 ± 2.0 | 73.0 ± 2.1 | 71.7 ± 2.2 | 60.4 ± 2.2 |
| | 128 | **77.3 ± 1.2** | 76.0 ± 1.1 | 73.7 ± 2.6 | 71.8 ± 2.7 | 63.4 ± 1.5 |
| Rec. $\hat{y}$ | 1 | **100.0 ± 0.0** | **100.0 ± 0.0** | **100.0 ± 0.0** | **100.0 ± 0.0** | 38.9 ± 14.6 |
| | 2 | 99.1 ± 6.0 | **99.3 ± 5.0** | 98.7 ± 7.1 | 95.9 ± 12.0 | 38.4 ± 11.5 |
| | 4 | **99.6 ± 3.0** | 99.0 ± 4.9 | 98.7 ± 6.3 | 96.8 ± 8.5 | 43.2 ± 7.2 |
| | 8 | **95.9 ± 7.8** | 95.3 ± 8.3 | 93.4 ± 7.2 | 91.9 ± 7.9 | 49.4 ± 4.6 |
| | 16 | **91.2 ± 7.3** | 89.1 ± 8.3 | 80.5 ± 4.7 | 77.4 ± 5.4 | 53.0 ± 3.1 |
| | 32 | **83.2 ± 4.1** | 80.9 ± 4.3 | 72.7 ± 2.2 | 71.0 ± 2.0 | 57.6 ± 2.3 |
| | 64 | **77.2 ± 2.4** | 76.0 ± 2.2 | 72.7 ± 2.1 | 71.5 ± 2.4 | 60.4 ± 2.2 |
| | 128 | **77.1 ± 1.2** | 75.9 ± 1.3 | 73.9 ± 2.7 | 71.8 ± 2.8 | 63.4 ± 1.5 |

Table 11: The mean inversion accuracy [%] and standard deviation of different methods over varying batch sizes with given true labels (top) and with reconstructed labels (bottom) on the **Health Heritage** dataset.

| Label | Batch Size | TabLeak | TabLeak (no pooling) | TabLeak (no softmax) | Cosine | Random |
|---|---|---|---|---|---|---|
| True $y$ | 1 | **99.8 ± 1.6** | **99.8 ± 1.6** | **99.8 ± 1.6** | **99.8 ± 1.6** | 34.8 ± 13.1 |
| | 2 | 97.7 ± 8.3 | 97.2 ± 10.2 | **98.6 ± 3.3** | 97.9 ± 5.6 | 36.9 ± 9.8 |
| | 4 | **98.2 ± 6.5** | 96.1 ± 9.8 | 97.8 ± 4.2 | 95.6 ± 8.1 | 37.0 ± 5.3 |
| | 8 | **96.0 ± 8.2** | 94.2 ± 10.5 | 89.2 ± 9.1 | 86.2 ± 9.0 | 39.2 ± 3.8 |
| | 16 | **86.1 ± 8.8** | 80.6 ± 9.9 | 67.8 ± 4.8 | 63.6 ± 5.5 | 41.4 ± 3.7 |
| | 32 | **70.0 ± 4.5** | 64.7 ± 3.9 | 61.4 ± 4.0 | 57.7 ± 4.1 | 43.4 ± 2.8 |
| | 64 | **64.7 ± 2.8** | 59.6 ± 2.7 | 61.5 ± 4.3 | 57.4 ± 4.7 | 45.0 ± 3.7 |
| | 128 | **63.0 ± 1.4** | 57.9 ± 1.6 | 59.9 ± 5.0 | 55.6 ± 4.8 | 46.8 ± 3.2 |
| Rec. $\hat{y}$ | 1 | 99.8 ± 1.6 | **99.9 ± 0.8** | 99.8 ± 1.6 | 99.6 ± 2.5 | 34.8 ± 13.1 |
| | 2 | **95.4 ± 13.6** | 94.8 ± 15.1 | 95.2 ± 13.6 | 92.5 ± 16.9 | 36.9 ± 9.8 |
| | 4 | **86.6 ± 20.2** | 84.7 ± 22.0 | 84.7 ± 20.8 | 83.5 ± 20.7 | 37.0 ± 5.3 |
| | 8 | **82.4 ± 15.6** | 80.5 ± 16.3 | 77.3 ± 13.3 | 74.5 ± 13.8 | 39.2 ± 3.8 |
| | 16 | **75.9 ± 12.4** | 71.4 ± 11.4 | 64.8 ± 7.6 | 60.9 ± 6.3 | 41.4 ± 3.7 |
| | 32 | **64.8 ± 5.7** | 60.8 ± 4.7 | 59.8 ± 3.8 | 56.9 ± 4.0 | 43.4 ± 2.8 |
| | 64 | **62.6 ± 2.6** | 59.6 ± 2.6 | 60.9 ± 4.0 | 57.7 ± 4.7 | 45.0 ± 3.7 |
| | 128 | **62.7 ± 1.6** | 59.2 ± 1.6 | 59.6 ± 5.1 | 55.7 ± 5.0 | 46.8 ± 3.2 |

Table 12: Mean and standard deviation of the inversion accuracy [%] with local dataset size of 32 in FedAvg training on the **Adult** dataset. The accuracy of the random baseline for 32 datapoints is $58.0 ± 2.9$.

| | TabLeak | | | Cosine | | |
|---|---|---|---|---|---|---|
| n. batches | 1 epoch | 5 epochs | 10 epochs | 1 epoch | 5 epochs | 10 epochs |
| 1 | **77.4 ± 4.5** | **71.1 ± 2.9** | **67.6 ± 3.7** | 65.2 ± 2.7 | 56.1 ± 4.1 | 53.2 ± 4.2 |
| 2 | **75.7 ± 5.0** | **71.7 ± 3.9** | **67.7 ± 4.2** | 64.8 ± 3.3 | 56.4 ± 4.8 | 56.2 ± 4.8 |
| 4 | **75.9 ± 4.4** | **71.0 ± 3.2** | **67.4 ± 3.4** | 64.8 ± 3.4 | 58.7 ± 4.6 | 56.6 ± 5.0 |

Table 13: Mean and standard deviation of the inversion accuracy [%] with local dataset size of 32 in FedAvg training on the **German Credit** dataset. The accuracy of the random baseline for 32 datapoints is $56.9 ± 2.1$.

| | TabLeak | | | Cosine | | |
|---|---|---|---|---|---|---|
| n. batches | 1 epoch | 5 epochs | 10 epochs | 1 epoch | 5 epochs | 10 epochs |
| 1 | **95.2 ± 3.8** | **87.9 ± 6.2** | **83.4 ± 4.6** | 78.2 ± 4.6 | 65.4 ± 6.2 | 62.5 ± 6.1 |
| 2 | **95.5 ± 3.9** | **88.2 ± 5.2** | **84.0 ± 6.6** | 78.3 ± 5.8 | 68.8 ± 6.6 | 63.4 ± 4.8 |
| 4 | **95.6 ± 3.6** | **85.5 ± 6.0** | **81.0 ± 6.1** | 79.2 ± 4.9 | 67.4 ± 4.8 | 62.6 ± 6.5 |

Table 14: Mean and standard deviation of the inversion accuracy [%] with local dataset size of 32 in FedAvg training on the **Lawschool Admissions** dataset. The accuracy of the random baseline for 32 datapoints is $57.8 ± 2.3$.

| | TabLeak | | | Cosine | | |
|---|---|---|---|---|---|---|
| n. batches | 1 epoch | 5 epochs | 10 epochs | 1 epoch | 5 epochs | 10 epochs |
| 1 | **85.6 ± 3.8** | **83.3 ± 2.9** | **80.7 ± 4.1** | 72.2 ± 2.6 | 68.1 ± 3.1 | 65.2 ± 2.8 |
| 2 | **86.0 ± 3.8** | **83.0 ± 3.2** | **79.8 ± 3.5** | 72.5 ± 1.9 | 68.3 ± 4.4 | 66.2 ± 2.8 |
| 4 | **85.8 ± 3.5** | **81.7 ± 3.8** | **79.3 ± 4.3** | 72.5 ± 2.4 | 69.4 ± 3.9 | 67.9 ± 3.8 |

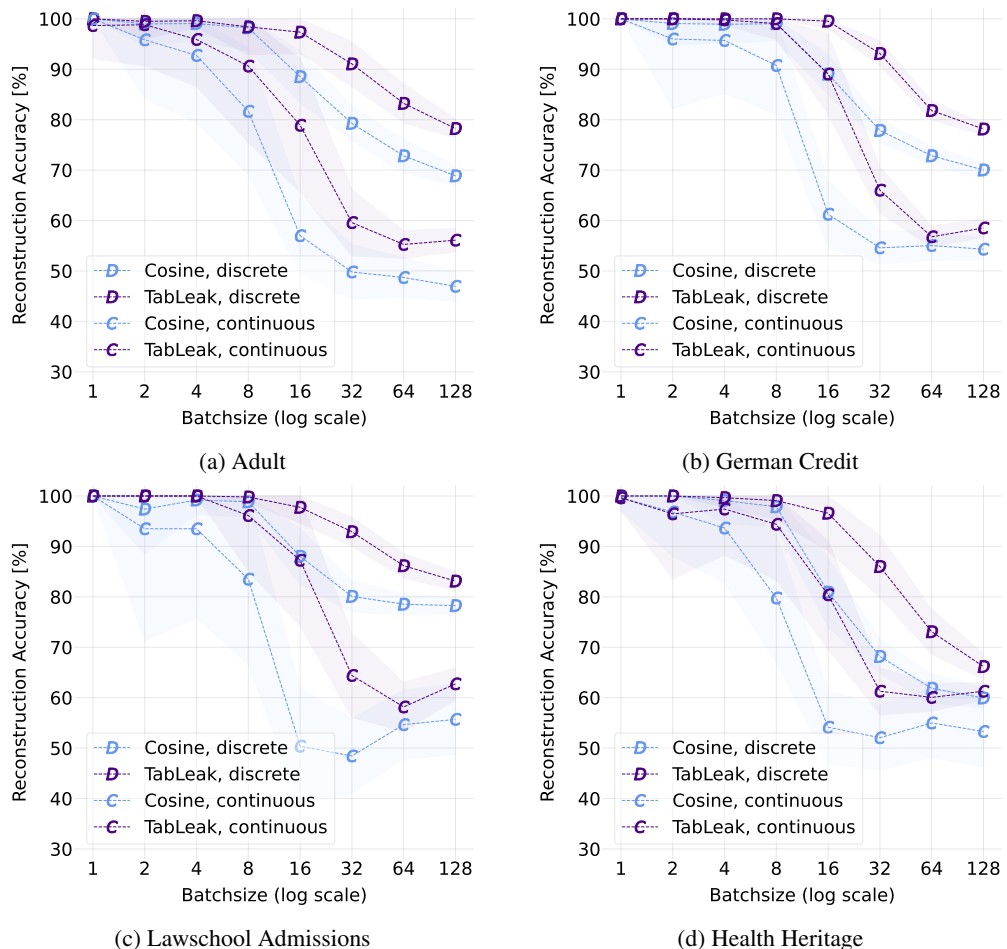

Figure 11: Mean reconstruction accuracy curves with corresponding standard deviations over varying batch size, separately for the discrete and the continuous features on all four datasets.

Table 15: Mean and standard deviation of the inversion accuracy [%] with local dataset size of 32 in FedAvg training on the **Health Heritage** dataset. The accuracy of the random baseline for 32 datapoints is $43.4 \pm 3.5$.

| | TabLeak | | | Cosine | | |
| n. batches | 1 epoch | 5 epochs | 10 epochs | 1 epoch | 5 epochs | 10 epochs |
|---|---|---|---|---|---|---|
| 1 | **68.5 ± 5.0** | **62.2 ± 3.5** | **57.4 ± 3.0** | 53.8 ± 5.5 | 41.4 ± 3.6 | 41.1 ± 3.4 |
| 2 | **68.1 ± 4.9** | **62.4 ± 4.1** | **57.0 ± 2.8** | 52.4 ± 5.7 | 43.4 ± 4.28 | 44.4 ± 4.3 |
| 4 | **67.3 ± 5.8** | **62.0 ± 3.5** | **57.0 ± 3.0** | 52.5 ± 6.6 | 43.4 ± 5.7 | 44.8 ± 4.4 |

Table 16: The mean accuracy [%] and entropies with the corresponding standard deviations over batch sizes of the categorical and the continuous features on the **Adult** dataset.

|  | Discrete | | Continuous | |
|---|---|---|---|---|
|  | Accuracy | Entropy | Accuracy | Entropy |
| 1 | $100.0 \pm 0.0$ | $0.01 \pm 0.04$ | $98.7 \pm 6.5$ | $-3.1 \pm 0.26$ |
| 2 | $99.5 \pm 3.5$ | $0.01 \pm 0.06$ | $98.8 \pm 8.2$ | $-2.82 \pm 0.57$ |
| 4 | $99.5 \pm 1.4$ | $0.08 \pm 0.11$ | $95.8 \pm 9.9$ | $-1.89 \pm 0.99$ |
| 8 | $98.5 \pm 5.6$ | $0.15 \pm 0.13$ | $90.9 \pm 14.7$ | $-1.11 \pm 0.95$ |
| 16 | $97.2 \pm 4.3$ | $0.26 \pm 0.11$ | $78.8 \pm 13.5$ | $-0.11 \pm 0.63$ |
| 32 | $91.0 \pm 4.4$ | $0.40 \pm 0.06$ | $59.2 \pm 6.9$ | $0.77 \pm 0.30$ |
| 64 | $83.2 \pm 3.6$ | $0.48 \pm 0.04$ | $55.1 \pm 3.0$ | $1.21 \pm 0.19$ |
| 128 | $78.5 \pm 1.8$ | $0.53 \pm 0.03$ | $55.7 \pm 2.0$ | $1.48 \pm 0.10$ |

Table 17: The mean accuracy [%] and entropies with the corresponding standard deviations over batch sizes of the categorical and the continuous features on the **German Credit** dataset.

|  | Discrete | | Continuous | |
|---|---|---|---|---|
|  | Accuracy | Entropy | Accuracy | Entropy |
| 1 | $100.0 \pm 0.0$ | $0.00 \pm 0.01$ | $100.0 \pm 0.0$ | $-3.10 \pm 0.18$ |
| 2 | $100.0 \pm 0.0$ | $0.03 \pm 0.05$ | $100.0 \pm 0.0$ | $-2.41 \pm 0.97$ |
| 4 | $100.0 \pm 0.0$ | $0.07 \pm 0.05$ | $99.8 \pm 1.1$ | $-1.66 \pm 0.80$ |
| 8 | $100.0 \pm 0.0$ | $0.10 \pm 0.07$ | $99.1 \pm 3.1$ | $-1.38 \pm 0.54$ |
| 16 | $99.5 \pm 1.4$ | $0.25 \pm 0.07$ | $89.1 \pm 8.1$ | $-0.35 \pm 0.22$ |
| 32 | $93.0 \pm 2.1$ | $0.43 \pm 0.04$ | $66.0 \pm 4.9$ | $0.60 \pm 0.13$ |
| 64 | $81.9 \pm 1.8$ | $0.56 \pm 0.02$ | $57.5 \pm 2.2$ | $1.08 \pm 0.06$ |
| 128 | $78.2 \pm 1.1$ | $0.59 \pm 0.02$ | $58.4 \pm 1.7$ | $1.30 \pm 0.05$ |

### D.4 FULL RESULTS ON ENTROPY ON ALL DATASETS

In Tab. 16, Tab. 17, Tab. 18, and Tab. 19 we provide the mean and standard deviation of the reconstruction accuracy and the entropy of the continuous and the categorical features over increasing batch size for attacking with TabLeak on the four datasets. In support of Sec. 4, we can observe on all datasets a trend of increasing entropy over decreasing reconstruction accuracy as the batch size is increased; and as such providing a signal to the attacker about their overall reconstruction success.

To generalize our results on the local information contained in the entropy, we show the mean reconstruction accuracy of both the discrete and the continuous features with respect to bucketing them based on their entropy in a batch of size 128 in Tab. 20, Tab. 21, Tab. 22, and Tab. 23 for all four datasets, respectively. We can see that with the help of this bucketing, we can identify subsets of the reconstructed features that have been retrieved with a (sometimes significantly *e.g.,* up to 24%) higher accuracy than the overall batch.

Table 18: The mean accuracy [%] and entropies with the corresponding standard deviations over batch sizes of the categorical and the continuous features on the **Lawschool Admissions** dataset.

|  | Discrete | | Continuous | |
|---|---|---|---|---|
|  | Accuracy | Entropy | Accuracy | Entropy |
| 1 | $100.0 \pm 0.0$ | $0.01 \pm 0.03$ | $100.0 \pm 0.0$ | $-3.28 \pm 0.29$ |
| 2 | $100.0 \pm 0.0$ | $0.02 \pm 0.05$ | $100.0 \pm 0.0$ | $-2.85 \pm 0.87$ |
| 4 | $100.0 \pm 0.0$ | $0.03 \pm 0.04$ | $100.0 \pm 0.0$ | $-2.45 \pm 0.78$ |
| 8 | $99.8 \pm 1.1$ | $0.11 \pm 0.10$ | $96.4 \pm 11.1$ | $-1.77 \pm 0.62$ |
| 16 | $98.1 \pm 2.8$ | $0.24 \pm 0.11$ | $87.1 \pm 13.4$ | $-0.65 \pm 0.49$ |
| 32 | $93.4 \pm 3.0$ | $0.41 \pm 0.08$ | $65.1 \pm 8.0$ | $0.18 \pm 0.20$ |
| 64 | $87.0 \pm 2.4$ | $0.55 \pm 0.05$ | $57.7 \pm 4.8$ | $0.78 \pm 0.12$ |
| 128 | $83.5 \pm 1.5$ | $0.60 \pm 0.03$ | $62.6 \pm 3.4$ | $1.07 \pm 0.11$ |

Table 19: The mean accuracy [%] and entropies with the corresponding standard deviations over batch sizes of the categorical and the continuous features on the **Health Heritage** dataset.

|  | Discrete | | Continuous | |
|---|---|---|---|---|
|  | Accuracy | Entropy | Accuracy | Entropy |
| 1 | $100.0 \pm 0.0$ | $0.02 \pm 0.05$ | $99.6 \pm 2.5$ | $-2.97 \pm 0.33$ |
| 2 | $100.0 \pm 0.0$ | $0.05 \pm 0.09$ | $96.5 \pm 12.9$ | $-2.55 \pm 0.80$ |
| 4 | $99.7 \pm 1.8$ | $0.08 \pm 0.10$ | $97.5 \pm 9.0$ | $-1.71 \pm 0.79$ |
| 8 | $99.1 \pm 3.7$ | $0.13 \pm 0.11$ | $94.3 \pm 11.3$ | $-1.06 \pm 0.64$ |
| 16 | $96.6 \pm 7.5$ | $0.26 \pm 0.10$ | $80.2 \pm 11.2$ | $-0.11 \pm 0.42$ |
| 32 | $85.1 \pm 6.4$ | $0.43 \pm 0.06$ | $61.7 \pm 3.8$ | $0.72 \pm 0.23$ |
| 64 | $73.1 \pm 4.7$ | $0.52 \pm 0.03$ | $59.6 \pm 2.5$ | $1.13 \pm 0.20$ |
| 128 | $66.1 \pm 2.4$ | $0.57 \pm 0.02$ | $60.7 \pm 1.6$ | $1.44 \pm 0.13$ |

Table 20: The mean accuracy [%] and the share of data [%] in each entropy bucket for batch size 128 on the **Adult** dataset.

| Entropy Bucket | Categorical Features | | Entropy Bucket | Continuous Features | |
|---|---|---|---|---|---|
|  | Accuracy [%] | Data [%] |  | Accuracy [%] | Data [%] |
| 0.0-0.2 | 95.7 | 8.1 | $\infty$-0.72 | 72.7 | 1.2 |
| 0.2-0.4 | 90.5 | 23.4 | 0.72-1.16 | 65.5 | 13.6 |
| 0.4-0.6 | 79.8 | 27.7 | 1.16-1.6 | 56.4 | 50 |
| 0.6-0.8 | 69.8 | 29.2 | 1.6-2.04 | 51.1 | 32.4 |
| 0.8-1.0 | 61.2 | 11.6 | 2.04-$\infty$ | 41.8 | 2.9 |
| Overall | 78.5 | 100 | Overall | 55.7 | 100 |
| Random | 73.8 | 100 | Random | 44.4 | 100 |

Table 21: The mean accuracy [%] and the share of data [%] in each entropy bucket for batch size 128 on the **German Credit** dataset.

| Entropy | Categorical Features | | Entropy | Continuous Features | |
|---|---|---|---|---|---|
| Bucket | Accuracy [%] | Data [%] | Bucket | Accuracy [%] | Data [%] |
| 0.0-0.2 | 98.1 | 7.4 | $\infty$-0.72 | 55.7 | 1.2 |
| 0.2-0.4 | 92.5 | 15.3 | 0.72-1.16 | 62.3 | 28.1 |
| 0.4-0.6 | 83.3 | 22.2 | 1.16-1.6 | 57.7 | 56.4 |
| 0.6-0.8 | 71.6 | 33.7 | 1.6-2.04 | 53.4 | 13.2 |
| 0.8-1.0 | 66.0 | 21.3 | 2.04-$\infty$ | 48.2 | 0.2 |
| Overall | 78.2 | 100 | Overall | 58.4 | 100 |
| Random | 73.5 | 100 | Random | 37.8 | 100 |

Table 22: The mean accuracy [%] and the share of data [%] in each entropy bucket for batch size 128 on the **Lawschool Admissions** dataset.

| Entropy | Categorical Features | | Entropy | Continuous Features | |
|---|---|---|---|---|---|
| Bucket | Accuracy [%] | Data [%] | Bucket | Accuracy [%] | Data [%] |
| 0.0-0.2 | 95.5 | 3.4 | $\infty$-0.72 | 69.5 | 20.7 |
| 0.2-0.4 | 92.1 | 14.2 | 0.72-1.16 | 63.3 | 35.1 |
| 0.4-0.6 | 88.0 | 32.7 | 1.16-1.6 | 60.1 | 32.9 |
| 0.6-0.8 | 81.2 | 30.4 | 1.6-2.04 | 55.5 | 10.8 |
| 0.8-1.0 | 70.7 | 19.3 | 2.04-$\infty$ | 54.1 | 0.5 |
| Overall | 83.5 | 100 | Overall | 62.6 | 100 |
| Random | 81.1 | 100 | Random | 19.1 | 100 |

Table 23: The mean accuracy [%] and the share of data [%] in each entropy bucket for batch size 128 on the **Health Heritage** dataset.

| Entropy | Categorical Features | | Entropy | Continuous Features | |
|---|---|---|---|---|---|
| Bucket | Accuracy [%] | Data [%] | Bucket | Accuracy [%] | Data [%] |
| 0.0-0.2 | 90.7 | 6.2 | $\infty$-0.72 | 69.1 | 1.1 |
| 0.2-0.4 | 84.7 | 22.1 | 0.72-1.16 | 65.6 | 17.0 |
| 0.4-0.6 | 70.5 | 21.2 | 1.16-1.6 | 61.8 | 52.9 |
| 0.6-0.8 | 54.8 | 32.3 | 1.6-2.04 | 55.9 | 26.5 |
| 0.8-1.0 | 50.3 | 18.4 | 2.04-$\infty$ | 49.4 | 2.5 |
| Overall | 66.1 | 100 | Overall | 60.7 | 100 |
| Random | 69.8 | 100 | Random | 34.2 | 100 |

# E    STUDYING POOLING

In this subsection, we present three further experiments on justifying and understanding our choices in pooling:

- Experiments on synthetic datasets for understanding the motivation for pooling in App. E.1.
- Ablation study on understanding the impact of the number of samples N on the performance of TabLeak in App. E.2.
- Comparison of using mean and median pooling on TabLeak in App. E.3.

## E.1    VARIANCE STUDY

A unique challenge (challenge (ii)) of tabular data leakage is that the mix of discrete and continuous features introduces further variance in the final reconstructions. As a solution to this challenge, we propose to produce $N$ independent reconstructions of the same batch, and ensemble them using the pooling scheme described in Sec. 3.2. In this subsection, we provide empirical evidence for the subject of challenge (ii) and the effectiveness of our proposed solution to it.

**Experimental Setup**    We create 6 synthetic binary classification datasets, each with 10 features, however of varying modality. Concretely; we have the following setups:

- Synthetic dataset with 0 discrete and 10 continuous columns,
- Synthetic dataset with 2 discrete and 8 continuous columns,
- Synthetic dataset with 4 discrete and 6 continuous columns,
- Synthetic dataset with 6 discrete and 4 continuous columns,
- Synthetic dataset with 8 discrete and 2 continuous columns,
- Synthetic dataset with 10 discrete and 0 continuous columns.

The continuous features are Gaussians with means between 0 and 5, and standard deviations between 1 and 3. The discrete features have domain sizes between 2 and 6, and the probabilities are drawn randomly. On each of these datasets we sample 50 batches of size 32 and reconstruct them using TabLeak (no pooling) starting from 30 different initializations in the same experimental setup elaborated in Sec. 4 and in App. B. We then proceed to calculate the standard deviation of the accuracy for each of the 50 batches over their 30 independent reconstructions, providing us 50 statistically independent data points for understanding the variance in the non-pooled reconstruction problem. Further, from the 30 independent reconstructions of each batch, we build 6 independent mini-ensembles of size 5 and conduct median pooling on them (essentially, TabLeak with $N = 5$). We then measure the standard deviation of the error for each of the 50 batches over the 6 obtained pooled reconstructions, obtaining 50 independent data points for analyzing the variance of pooled reconstruction.

**Results**    We present the results of the experiment in Fig. 12; additionally to measuring the same-batch reconstruction accuracy standard deviation for all features together, we also present the resulting measurements when only considering the discrete and the continuous features, respectively. The figures are organized such that the x-axis begins with the synthetic dataset consisting only of continuous features and progresses to the right by decreasing the number of continuous and increasing the number of discrete features at each step by 2. Roughly speaking, the very left column of the figures is similar to data leakage in the image domain, where all features are continuous, and the very right relates to data leakage in the text domain, containing only discrete features. Looking at Fig. 12a, we observe that the mean same-batch STD is indeed higher for datasets consisting of mixed types, providing empirical evidence underlining the second challenge of tabular data leakage. Further, it can be clearly seen that pooling, even with a small ensemble of just 5 samples, decisively decreases the variance of the reconstruction problem, providing strong justification for using pooling in the tabular setting. Finally, from Fig. 12b and Fig. 12c we gain interesting insight in the underlying dynamics of the interplay between discrete and continuous features at reconstruction. Concretely, we observe that as the presence of a given modality is decreasing and its place is taken up by the other, the recovery

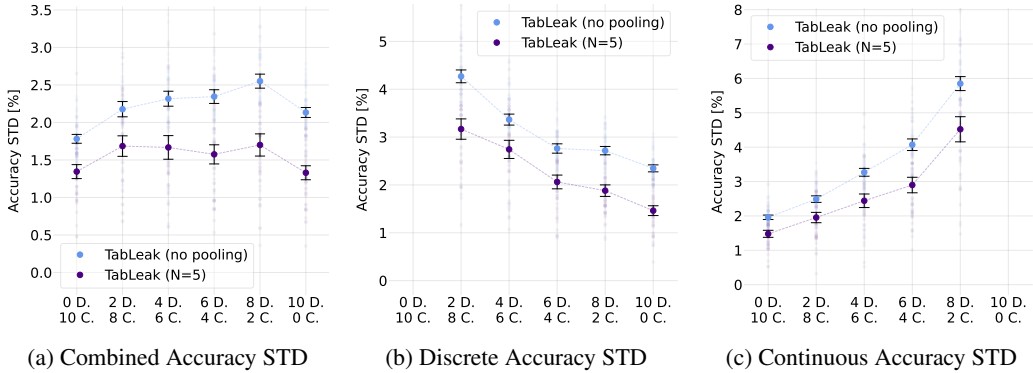

(a) Combined Accuracy STD    (b) Discrete Accuracy STD    (c) Continuous Accuracy STD

Figure 12: Mean same-batch reconstruction accuracy standard deviation and 90% confidence interval at batch size 32 estimated from 50 independent batches over synthetic datasets with varying number of discrete (D.) and continuous (C.) features.

Table 24: Reconstruction accuracy [%] and standard deviation of TabLeak on batch size 32 over the size of the ensemble $N$ used for pooling.

| $N =$ | 1 | 5 | 10 | 15 | 20 | 25 | 30 |
|---|---|---|---|---|---|---|---|
| Adult | $71.8 \pm 4.6$ | $75.1 \pm 4.6$ | $76.5 \pm 4.6$ | $76.5 \pm 4.8$ | $77.1 \pm 4.7$ | $77.0 \pm 4.7$ | $\mathbf{77.4 \pm 4.8}$ |
| German | $79.0 \pm 3.0$ | $81.6 \pm 2.9$ | $82.6 \pm 2.9$ | $82.9 \pm 2.9$ | $83.2 \pm 2.8$ | $83.4 \pm 2.8$ | $\mathbf{83.6 \pm 2.7}$ |
| Lawschool | $82.3 \pm 4.0$ | $84.4 \pm 3.7$ | $84.7 \pm 4.1$ | $85.2 \pm 3.9$ | $85.1 \pm 3.9$ | $\mathbf{85.3 \pm 4.0}$ | $\mathbf{85.3 \pm 4.0}$ |
| Health Heritage | $64.6 \pm 4.2$ | $67.5 \pm 4.3$ | $69.1 \pm 4.4$ | $69.1 \pm 4.5$ | $69.7 \pm 4.5$ | $69.5 \pm 4.3$ | $\mathbf{70.1 \pm 4.0}$ |

of this modality becomes increasingly noisier. Much in line with the observations on the difference in the recovery success between discrete and continuous features, these results also argue for future work to pursue methods that decrease the disparity between the two different feature types in the mixed setting.

### E.2    THE IMPACT OF THE NUMBER OF SAMPLES N

In Tab. 24 we present the results of an ablation study we conducted on TabLeak at batch size 32 to understand the impact of the size of the ensemble $N$ on the performance of the attack. We observed that with increasing N the performance of the attack gets steadily better, albeit, producing diminishing returns, showing signs of saturation on some datasets after $N = 25$. Note that this behavior is expected, and suggests using the largest $N$ that is not yet computationally prohibitive. We chose $N = 30$ for all our experiments with TabLeak (unless explicitly stated otherwise); this allowed us to conduct large-scale experiments while still extracting good performance from TabLeak.

### E.3    CHOICE OF THE POOLING FUNCTION

We compare TabLeak using median pooling to TabLeak with mean pooling in Tab. 25 over the four datasets. As we can observe, in most cases both methods produce similar results, hence the effectiveness of TabLeak is not to be attributed solely to the chosen pooling method. However, as median pooling demonstrates to provide a slight edge in some cases, we opt for using median pooling in our main experiments with TabLeak.

| batch size | TabLeak (median) | TabLeak (mean) |
|---|---|---|
| 1 | **99.4 ± 2.8** | **99.4 ± 2.8** |
| 2 | 99.2 ± 5.5 | **99.3 ± 5.0** |
| 4 | **98.0 ± 4.5** | 97.7 ± 5.3 |
| 8 | **95.1 ± 9.2** | 94.8 ± 9.0 |
| 16 | **89.4 ± 7.6** | 88.9 ± 7.7 |
| 32 | **77.6 ± 4.8** | 77.1 ± 4.7 |
| 64 | 71.2 ± 2.8 | **71.7 ± 2.8** |
| 128 | 68.8 ± 1.3 | **69.4 ± 1.4** |

(a) Adult

| batch size | TabLeak (median) | TabLeak (mean) |
|---|---|---|
| 1 | **100.0 ± 0.0** | **100.0 ± 0.0** |
| 2 | **100.0 ± 0.0** | **100.0 ± 0.0** |
| 4 | **99.9 ± 0.4** | **99.9 ± 0.4** |
| 8 | **99.7 ± 1.1** | 99.6 ± 1.1 |
| 16 | **95.9 ± 3.4** | 95.6 ± 3.3 |
| 32 | **83.6 ± 2.9** | 83.1 ± 3.0 |
| 64 | **73.0 ± 1.3** | 72.6 ± 1.3 |
| 128 | **71.3 ± 0.8** | 70.8 ± 0.9 |

(b) German Credit

| batch size | TabLeak (median) | TabLeak (mean) |
|---|---|---|
| 1 | **100.0 ± 0.0** | **100.0 ± 0.0** |
| 2 | **100.0 ± 0.0** | **100.0 ± 0.0** |
| 4 | **100.0 ± 0.0** | **100.0 ± 0.0** |
| 8 | 98.7 ± 3.8 | **98.8 ± 3.4** |
| 16 | **94.8 ± 5.6** | 94.6 ± 5.4 |
| 32 | **84.8 ± 3.9** | 84.7 ± 3.9 |
| 64 | **78.2 ± 2.0** | **78.2 ± 2.2** |
| 128 | 77.3 ± 1.2 | **77.5 ± 1.2** |

(c) Lawschool Admissions

| batch size | TabLeak (median) | TabLeak (mean) |
|---|---|---|
| 1 | **99.8 ± 1.6** | **99.8 ± 1.6** |
| 2 | **97.7 ± 8.3** | 97.4 ± 9.0 |
| 4 | **98.2 ± 6.5** | 98.0 ± 6.7 |
| 8 | **96.0 ± 8.2** | 95.6 ± 8.6 |
| 16 | **86.1 ± 8.8** | 84.9 ± 9.3 |
| 32 | **70.0 ± 4.5** | 69.7 ± 4.4 |
| 64 | 64.7 ± 2.8 | **64.8 ± 2.9** |
| 128 | 63.0 ± 1.4 | **63.6 ± 1.5** |

(d) Health Heritage

Table 25: Mean and standard deviation of the inversion accuracy [%] using TabLeak with either median or mean pooling, assuming full knowledge of the true labels.

