# OpenReview forum: "Data Leakage in Tabular Federated Learning"
_ICLR.cc/2023/Conference — Submitted to ICLR 2023_

### Official Review · Reviewer_Gtmh · 2022-10-23

**Confidence:** 4
**Correctness:** 3
**Technical Novelty And Significance:** 3
**Empirical Novelty And Significance:** 4
**Recommendation:** 6

**Clarity, Quality, Novelty And Reproducibility:**

# Clarity

- The paper is well written and easy to follow.

## Minor typos and comments

- I would recommend to standardize notations across the paper: the symbols $\mathcal{D}_i$ is used to denote private datasets at the beginning of section 2, and discrete sets of features at the end of the same section. Similarly, $\sigma$ denotes the softmax function in the main paper and the variance of the Gaussian Differential privacy noise in Appendix C.1. Also, index $i$ is used for different purposes
- In the introduction of "mixed type tabular data" (section 2), I think there is some confusion in the sentence introducing "the one-hot encoding of the $i$-th discrete...". I suspect it should read "The one-hot encoding of the $i$-th discrete feature $x_i$ is a vector $c_i^D(x)$
- Page 4, notations for $c(x)$ are not the same at the top (continuous features marked $x_i^C$) and at the bottom (continuous marked $z_i^C$)
- Sec 3.3, paragraph "categorical features", one should have $\hat{x}_{ij}^D$ and not C in the exponent
- sec 3.3, end of first line, "to asses" -> "to assess"
- In table 2, the first column is entitled #batches but not entirely clear to me. Is it the batch size or the number of batches per epoch?

# Quality

- I find the experimental demonstration of very good quality overall. I have several comments, though.
- One claim of the paper is that the mix of continuous and discrete features creates a harder optimization problem (with higher variance), thereby justifying the softmax+pooling approaches.
  - With the current results, I do not see exactly where one sees that it is the very /mixture/ of discrete and continuous features that makes the problem difficult. I would be more convinced e.g. if there was a link between the prevalence of continuous and discrete variables among total variables and the final accuracy of the baseline method. In the datasets, what is the % of discrete vs continuous variables?
  - Further, I do not see evidence that the same mix leads to a higher variance than fully continuous problems, as there is no such comparison.
  - However, I agree that the softmax approach makes sense for this problem, and pooling indeed
- Regarding the discrete features, similarly to NLP, one have tried to leverage the discrete nature of data by looking at non-zero entries in the gradients or updates to know which categorical features are represented, and maybe constraining the optimization problem this way. Have the authors tested such an approach?
- Regarding pooling:
  - The problem it is supposed to tackle is not entirely justified, see above
  - What is the effect of $N$ (number of replicas) on the results?
  - What is the justification for a median pooling? In particular, there is no guarantee that a median pooling leads to features summing up to 1 (while a mean pooling would guarantee it) - how would mean pooling perform?
- The authors assume a gaussian distribution for continuous variables to compute the entropy, but it does not hold in practice, witness the variations of the random baselines for continuous variables in Tables 18-21: while the buckets of the accuracy metric are supposed to contain 25% of the probability mass under a gaussian hypothesis, the performance of the random (gaussian) baseline varies between 44.4% to 19.1%. How would you propose to extend the measure of the entropy in non-gaussian cases?
- In sec 4, baselines, the paper states
> the random baseline monotonously approach perfect accuracy with increasing batch size

  I don't understand how this can be possible with the definition of the random baseline, unless the data generating distribution corresponds to an independent product of marginals?
- Regarding the effect of the size of the neural network,
  - it would be more useful to compare the absolute numbers provided (e.g. 50 vs 100 hidden units...) to the number of features in the dataset, to understand the regime in which one is (underparametrized vs overparametrized).
  - a usually strong baseline for tabular data is a linear model (no hidden layer), which is not tackled here

# Novelty

- The application of gradient attacks to tabular data is novel to the best of my knowledge
- The use of pooling and softmax is novel, as well as the heuristic of the entropy
- For key challenge (i) mentioned by the authors in the introduction, the authors contrast the mixed discrete-continuous problem to the fully continuous optimization for pixels and embeddings for text. While I agree for pixels, I don't think it is fair to judge embeddings for texts as continuous. In practice, the embeddings are fixed for fixed network weights (which is the setting of the attack for FedSGD). In (Gupta et al., Recovering private text in federated learning of language models) cited by the authors, the attack precisely leverage the discrete nature of embeddings by looking at non-zero entries on the vector to find a bag of words present in the original data.

# Reproducibility

- I want to thank the authors for providing most experimental details.
- One important aspect that I could not find is the computation of the accuracy over batches: how do the authors align the reconstructed batch with the original one? e.g. do they follow the same matching algorithm as in Sec 3.2?
- In the experimental setup, 50 different batches are attacked for a given initialization of a network. Are there repetitions over the particular initializations of this network as well?
- For the cosine baseline on the discrete labels, did you add any constraints of discrete features summing up to 1?


**Details Of Ethics Concerns:**

No ethical concern

**Strength And Weaknesses:**

# Main strengths

- Simple method, easy to plug over existing approaches (Geiping et al.)
- Proposes a working entropy-based heuristic to assess the quality of reconstruction
- Compelling experimental demonstration for the effectiveness of the proposed method. I have several comments to improve it further, but I already find it very convincing

# Main weaknesses

- Some claims regarding the specificity of problem are a bit strong (see novelty below).
  - The related works on NLP are not completely fairly represented (see below);
  - The authors list as a challenge specific to tabular data that "the mix of categorical and continuous features causes high variance in the final reconstructions", but there is no empirical proof of it. However, there is no denying it brings improvements, and it also enables the entropy heuristic; it is mostly a wording issue. (see detailed comments)
- The justification and study of pooling could be improved (see below)

**Summary Of The Paper:**

This paper investigates the problem of data reconstruction from gradients in federated learning in the case of tabular data. Two main issues arise with this modality: some features are categorical, and it is much harder for a naïve human to guess whether the samples are correctly reconstructed.

Building on the previous work of Geiping (for images), the paper introduces a softmax parametrization and the resulting method optimizes on the resulting latent variables. Further, by using multiple independent reconstruction and subsequent pooling, reconstruction results are improved. The authors propose to leverage these independent reconstructions to measure the entropy of reconstructed features as a proxy for quality of reconstruction.

Many experiments on 4 tabular datasets (1 in main body, the remaining three in appendix) investigate in detail the proposed method. The authors first perform an ablation study to show the effect of each proposed improvement (softmax and pooling) in the case of FedSGD, for varying batch size, demonstrating a significant improvement over the baseline. The authors observe that discrete features are easier to retrieve than continuous ones. For FedAvg-style updates, experiments also demonstrate an improvement over baseline. The authors show empirically that the heuristic based on entropy is associated with a larger retrieval success for categorical variables.

**Summary Of The Review:**

The authors propose a novel approach to make gradient attacks work successfully on tabular data. The thorough experiments are enough to convince me of the viability of the approach. However, I have small concerns regarding the claims on the specificity of the problems tackled for tabular data. If the authors address them, I will raise my score accordingly.

---

> ### Author Response · Authors · 2022-11-14
> **Response Part 4/4**
>
> **Q12: Do you also take samples over different initializations of the network in your experiments?**
>
> We did not explicitly incorporate the network initializations into our experiments as we observed in the initial stage of experimentation that it has less of an impact on the inversion success than the sampled batch itself. However, over different experiments and different baselines, we did use different initializations of the network, suggesting that the global trends observed in the experiments are not dependent on the network initialization. To justify our claims, we conducted an experiment on TabLeak (no pooling); we sampled 10 batches of size 32 from the Adult dataset and reconstructed them each on 10 different initializations of the network. The resulting overall estimated performance is $72.3 \pm 4.8$ compared to $72.4 \pm 4.6$ when only considering different batches over the same initialization of the network as reported in the paper. We repeated this experiment also for the Cosine baseline, obtaining $66.6 \pm 3.3$ considering network initializations, compared to $66.6 \pm 3.5$ reported in the paper. Therefore, the impact of different network initializations on the final estimates is not significant, ultimately, allowing for better scalability of the experiments.
>
> **Q13: Did you conduct an experiment constraining the categorical features of the Cosine baseline to sum to 1?**
>
> We experimented with such approaches, mainly using a regularizer of the form $\lambda \mathcal{R} = \lambda \sum_{i=1}^K \left ( 1 - g(z_i^D) \right )^2$, where to accommodate for your suggestion, we would choose $g(z_i^D) = \sum_{j=1}^{D_i} z_{ij}^D$. However, the best performing choice for $g$ was the L1 norm, on which after tuning $\lambda$, we obtained $70.4 \pm 3.3$ on Adult at batch size 32, compared to $72.4 \pm 4.6$ of TabLeak (no pooling) and $66 \pm 3.5$ of the unconstrained Cosine Baseline; with the softmax remaining the better method. Also, in contrast to the softmax relaxation, such regularizers require the tuning of a hyperparameter, introducing additional complexity to the problem, while it is also not clear what the best practice for hyperparameter tuning in real-life scenarios would be (no access to data). Further, the softmax is a principled continuous relaxation of discrete variables, which we also observed to behave well during optimization and produce reconstructed encodings to resemble one-hot vectors. For these reasons we decided to pursue the softmax relaxation.
>
> **Q14: In the tables showing the experiments attacking FedAVG, does \#batches denote the number of batches used in an epoch?**
>
> Yes, we have updated the tables to say n. batches to avoid future confusion.
>
> **References**
>
> [1] Gupta, Samyak, et al., “Recovering Private Text in Federated Learning of Language Models”,  Advances in Neural Information Processing Systems. 2022.

---

> ### Author Response · Authors · 2022-11-14
> **Response Part 3/4**
>
> **Q6: Is it a problem that after median pooling the resulting encodings for the discrete features are not guaranteed to sum to one anymore?**
>
> No, because immediately after pooling we project the resulting reconstruction by taking the argmax over the resulting encodings of the discrete features. Hence, at this point we are not interested anymore in recovering vectors respecting the (relaxed) one-hot constraints, merely, we are interested in the ‘winner’. For the entropy calculation we use the relative frequency of the projected features in the ensemble.
>
> **Q7: The study of the impact of the network size could be improved, it is hard to understand the significance of the sheer number of parameters in the current presentation. Also, it would make sense to consider a simple linear model as well.**
>
> We have revised the corresponding section in the Appendix incorporating your suggestions. We also added a table in Appendix B with the full specifications of each dataset. In detail, we made the following changes to the section studying the impact of the network size (Appendix C.2): we added the simple linear model to the set of examined models, we report under each figure the number of features in the encoded input, and we trained all examined models for 50 epochs on all datasets and provided their maximum test accuracy over the whole training and their full test accuracy over training curves. The latter addition is meant to serve as a demonstration for the behavior of the models during training, where from their tendency to overfit or underperform it can be understood if the model is over- or under-parameterized. We observe in these experiments that while it is hard to attack the linear model, it performs worse than the non-linear neural networks in terms of accuracy. Also, the much larger networks that are very easy to attack also prove to be very prone to overfitting, hence they are strongly overparameterized for the underlying task. For the full experimental details, data, and conclusion, please see the updated Appendix C.2.
>
> **Q8: Can the random baseline in its current form reach perfect accuracy in the limit of the batch size?**
>
> In theory no, indeed, as correctly remarked by you this would only be the case if all features in the original dataset were independent. We have clarified this point in the revised version. It is important to remark, that even this naive baseline is improving in accuracy as the batch size grows, as it represents sampling from a similar distribution to the data generating one.
>
> **Q9: Why does the random baseline under the bucketing scheme in Tables 18-21 not present a uniform accuracy distribution despite the Gaussian hypothesis?**
>
> The bucketing in the mentioned tables does not correspond to the error-bounds used for calculating the accuracy. Merely, we chose those intervals in Tables 18-21 for the sake of presentation, with the goal of having a somewhat balanced distribution of the data over the buckets. However, the key message to be conveyed there is the same as with the categorical features; those features that exhibit low reconstruction entropy tend to produce also higher accuracy.
>
> **Q10: How can the entropy measure for the continuous features be extended beyond the Gaussian hypothesis?**
>
> The entropy measurement of the reconstructed continuous features is not defined by the distribution of the features in the original data generating distribution. Rather, we assume that at inversion, the inversion error behaves as Gaussian (as we assume for the accuracy metric, as you correctly pointed out) and measure the standard deviation of the features. Now, this is a mere heuristic that allows us to simply formalize capturing the uncertainty in the reconstructed features. This approach is clearly motivated by intuition, as a higher variance in the reconstructed features points to more uncertainty, and the entropy of a Gaussian random variable is in fact a monotonously increasing function of the variance. As such, the simplest alternative metric for capturing the uncertainty in the reconstruction would be just the measured variance of the reconstructed features. One could also possibly consider a discretization of the continuous features, and measuring the entropy of the resulting histogram. We did not test such approaches as we considered the current method sufficiently informative and simple.
>
> **Q11: How do you calculate the final accuracy with respect to the ground truth? Do you use a similar matching as for pooling?**
>
> Yes, in line with the methods established in prior works, we reorder the obtained reconstructions with respect to the ground truth through a maximum accuracy matching. Note that this is only necessary for the measurement step when reporting experimental results. When mounting the attack in practice it is not necessary to know the ‘true’ order of the lines in the batch.

---

> ### Author Response · Authors · 2022-11-14
> **Response Part 2/4**
>
> **Q3: Does the ‘mix’ of continuous and discrete features really result in an increased variance in the reconstruction problem, necessitating a variance reduction method like pooling?**
>
> Yes, the concurrent presence of discrete and continuous features in the data indeed increases the variance in the reconstruction problem. However, we agree that in the initial version of the paper we did not provide sufficient evidence for this claim, and we would like to thank you for pointing this out, as it led to an insightful experiment. We added a section in the Appendix (section Appendix E.1) studying how the mix of the two modalities impact the variance of the reconstruction.
>
> For this, we created 6 synthetic datasets with 10 features each, but with varying mixes of the modalities. Concretely, we created a dataset containing only continuous columns, a dataset containing only discrete columns, and four datasets ‘interpolating’ between them by 2 features at a time (2 discrete and 8 continuous, 4 discrete and 6 continuous, and so on). We then sampled 50 batches of size 32 from each dataset and reconstructed each of them 30 times starting from different initializations using TabLeak (no pooling), and recorded the mean of the reconstruction accuracy’s standard deviation for each batch over the 30 trials. Finally, we calculated the mean same-batch reconstruction StdDev over the 50 batches for each dataset. We then repeated the same experiment, but instead of measuring the reconstruction error on each of the 30 reconstructions for a given batch separately, we first built 6 mini-ensembles of size 5, on which we conducted median pooling (essentially TabLeak with $N=$5). In the table below we report the resulting mean same-batch reconstruction StdDevs ($\pm 90$\% confidence interval) over all 6 datasets (D=number of discrete columns, C=number of continuous columns), for both the non-pooled and the mini-ensembled attacks:
>
> |                       |D=0, C=10|D=2, C=8|D=4, C=6|D=6, C=4|D=8, C=2|D=10, C=0|
> |:----------------------|:-------:|:------:|:------:|:------:|:------:|:-------:|
> |TabLeak (no pooling)   |1.8 $\pm$ 0.1|2.2 $\pm$ 0.1|2.3 $\pm$ 0.1|2.3  $\pm$ 0.1|2.6 $\pm$ 0.1|2.1 $\pm$ 0.1|
> |TabLeak ($N=5$)        |1.3 $\pm$ 0.1|1.7 $\pm$ 0.1|1.7 $\pm$ 0.2|1.6 $\pm$ 0.1|1.7 $\pm$ 0.1|1.3 $\pm$ 0.1|
>
> From this table we can make two important observations: (i) the variance of the reconstruction problem is clearly higher on the mixed datasets, especially on the ones with a lot of discrete features (second to last column), (ii) pooling, even with as little samples as 5, has a significant impact on reducing the variance in the reconstruction problem. These two observations both validate our claim about the second challenge of tabular data leakage, and confirm the effectiveness of pooling mitigating exactly this issue. For further details and discussion of this experiment, please see Appendix E.1.
>
> **Q4: What is the impact on the results of the size of the ensemble $N$ in TabLeak?**
>
> We conducted an ablation study on the impact of $N$ on TabLeak. On each of the four datasets we examine $N$ on a scale from 1 to 30 at steps of five. The full results and details of the experiment are included now in Appendix E.2. In the following table we present the partial results, when looking at N at steps of 10:
>
> | $N =$           |    1         |    10        |    20        |    30        |
> |:--------------|:------------:|:------------:|:------------:|:------------:|
> |Adult          |71.8 $\pm$ 4.6|76.5 $\pm$ 4.6|77.1 $\pm$ 4.7|**77.4** $\pm$ **4.8**|
> |German         |79.0 $\pm$ 3.0|82.6 $\pm$ 2.9|83.2 $\pm$ 2.8|**83.6** $\pm$ **2.7**|
> |Lawschool      |82.3 $\pm$ 4.0|84.7 $\pm$ 4.1|85.1 $\pm$ 3.9|**85.3** $\pm$ **4.0**|
> |Health Heritage|64.6 $\pm$ 4.2|69.1 $\pm$ 4.4|69.7 $\pm$ 4.5|**70.1** $\pm$ **4.0**|
>
> We can observe from this table that the biggest accuracy gain is made by the jump from having no ensembling, to having ensembling with some number of samples. As we increase N, we observe increasing reconstruction accuracy, albeit at diminishing returns. For extracting the maximum performance, it is nevertheless worth it to run at the maximum number of samples computationally permitted. We opted for an ensemble size of 30, as we initially observed strong results, while still allowing us to efficiently scale our experiments.
>
> **Q5: What is the justification for median over mean pooling in TabLeak?**
>
> We observed that median pooling provides slightly better results than mean pooling. We included an experiment demonstrating this in Appendix E.3.

---

> ### Author Response · Authors · 2022-11-14
> **Response Part 1/4**
>
> First of all, we would like to sincerely thank you for your extensive and detailed review. We highly appreciate your inputs, and believe they helped increase the quality of our paper. Further, we are glad that you recognised the simplicity and the portability of our method; we believe that this aspect of TabLeak highlights the high vulnerability of FL on tabular data. We are also thankful for your recognition of the high empirical novelty of our work, acknowledging our thorough experimental evaluations and strong results. Further, thank you for pointing out minor typos and notational inconsistencies, we addressed them carefully.
>
> Below we address your questions and comments:
>
> **Q1: Could a similar technique of identifying the discrete features directly from the gradients, as used by Gupta et al. [1], work in this setting as well?**
>
> No, we explored this direction, but due to some important differences between the setting in [1] and our setting, mixed type tabular data, we believe it is not possible. In contrast to the setting in [1], we standardize all columns of the input, resulting in encodings of the discrete features that are not anymore informative about the inclusion of categories based solely on the zeros in the vector. We do so following common practice in data science, where we want all feature columns to be on the same scale to accommodate regularization methods in training; moreover, we observed that it makes the training of the networks more stable and results in a higher final accuracy. Crucially, the standardization can not be reversed from the gradients due to the averaging that happens when computing the batch gradient. Further, even if we were not to standardize the columns, in the tabular setting we apply ReLU activations to the outputs of the first linear layer which is a standard practice in most neural networks except for the token-to-embedding layer of text-specific transformers used in [1]. This has the practical implication that the gradients of some input categories are zeroed out by the ReLUs, which makes the attack of [1] wrongly exclude categories otherwise present in the input batch. Additionally, deeper-lying ReLU units result in further zero-masking of certain categories. In our experiments, these effects result in the false exclusion of at least one category 72\%-86\% of the time depending on the batch size. Finally, unlike text, discrete features in tabular data usually come from comparably small “vocabularies”. As the attack presented in [1] can only recover the presence of a category but not how many times it is present in a batch, the utility of the attack is severely limited in our setting. In our experiments we observed, that even if we assume that there are no false exclusions (which as we have seen above, strongly does not hold) and no standardization is used, we can exclude a meaningful portion of the categories only for smaller batch sizes (<32), on which we already achieve near perfect reconstruction of the categorical features with our current methods.
>
> **Q2: Data leakage attacks on text data can not be considered as fully continuous optimization problems.**
>
> Indeed, we agree, and have updated the introduction to be more clear about this point. As you have correctly stated, considering fixed embeddings of the tokens, data leakage in the text domain is inherently a fully discrete problem. Meanwhile, leakage in the image domain is fully continuous. Hence, data leakage in the tabular domain distinguishes itself from these other domains by lying ‘between’ them, consisting of both continuous and discrete columns, posing a unique challenge.

---

> ### Author Response · Authors · 2022-11-25
> **Follow-up**
>
> We would like to thank you once again for your thorough review, many helpful comments, and interesting questions raised. We would like to ask you to, please, review our responses, and consider reassessing our work. If you have any questions, comments, or concerns left, please, let us know, we are eager to engage in further discussion.

---

### Official Review · Reviewer_x5xg · 2022-10-23

**Confidence:** 5
**Correctness:** 4
**Technical Novelty And Significance:** 1
**Empirical Novelty And Significance:** 1
**Recommendation:** 3

**Clarity, Quality, Novelty And Reproducibility:**

The paper is well written. Perhaps the background section is too long and should be shortened.
The source code is available and the experiments can be reproduced.

**Strength And Weaknesses:**

Strengths:
The identification of the unique aspect of data reconstruction in the tabular setting

Weaknesses:
The underlying new ideas are not innovative. Softmax relaxations are well known and ensemble techniques are also ubiquitous.
The benchmark models are fairly weak (random selection and a technique transferred from the continuous case)



**Summary Of The Paper:**

While reconstructing data in FL from gradients leads to a continuous optimization problem, tabular data with categorical variables yields a mixed integer programming optimization problem. The authors propose a softmax based continuous relaxation and an ensemble strategy that makes the underlying solution algorithm more robust. They show 10% improvement over baselines.

**Summary Of The Review:**

Both technical contributions are weak. In terms of softmax, there are more sophisticated and often efficient relaxations of discrete random variables (for example, the Gumble trick).
It is also possible to tackle directly the mixed integer formulation. There is a slew of algorithms in the optimization community (for example, branch-and-bound, GA) that are applicable in this case (many of them don't require convexity and can handle general optimization problems).
Robustness can also be improved by not simply taking ensemble. There is work in distributed computing on how to effectively explore parallelization.

---

> ### Author Response · Authors · 2022-11-14
> **Response Part 3/3 - References**
>
> $\newcommand{Ro}{\textcolor{purple}{oUHx}}$
> $\newcommand{Rt}{\textcolor{green}{x5xg}}$
> $\newcommand{Rtr}{\textcolor{blue}{Gtmh}}$
>
> **References**
>
> [1] Jang, Eric, et al. "Categorical Reparametrization with Gumbel-Softmax", International Conference on Learning Representations. 2017.
> [2] Geiping, Jonas, et al. “Inverting Gradients - How easy is it to break privacy in federated learning?”, Advances in Neural Information Processing Systems. 2020.
> [3] Yin, Hongxu, et al. "See through gradients: Image batch recovery via gradinversion." Proceedings of the IEEE/CVF Conference on Computer Vision and Pattern Recognition. 2021.
> [4] Lu, Jiahao, et al. "APRIL: Finding the Achilles' Heel on Privacy for Vision Transformers," 2022 IEEE/CVF Conference on Computer Vision and Pattern Recognition (CVPR). 2022.
> [5] Balunovic, Mislav, et al. ”LAMP: Extracting Text from Gradients with Language Model Priors”, Advances in Neural Information Processing Systems. 2022.
> [6] Dimitrov, I. Dimitar, et al. “Data Leakage in Federated Averaging”, Transactions on Machine Learning Research. 2022.

---

> ### Author Response · Authors · 2022-11-14
> **Response Part 2/3**
>
> $\newcommand{Ro}{\textcolor{purple}{oUHx}}$
> $\newcommand{Rt}{\textcolor{green}{x5xg}}$
> $\newcommand{Rtr}{\textcolor{blue}{Gtmh}}$
>
> **Q3: Are the used baselines, random guessing and the Cosine baseline, necessary and sufficient for evaluating TabLeak?**
>
> Yes, we believe that the baselines used in our work to evaluate our method provide sufficient comparison for understanding and judging our results. (i) It is imperative to evaluate against the random baseline, because, as stated in the paper, any reconstruction method that does not outperform the random guessing is not extracting any useful information from the gradient. This baseline is especially needed for tabular data, as we observe that at larger batch sizes even random guessing achieves high accuracy. As such, it is non-trivial to beat it, and without this comparison the attack results could be deceptive. A good example for this is Table 2 in the paper, where one could falsely think that the Cosine baseline recovers a decent percentage of around 56% of all features even in the hardest FedAVG setting, however as this is below the random guessing baseline of 58%, we can not be sure that there is any input-specific information recovered by the attack (no information gain). (ii) The Cosine baseline is based on the strong attack developed in [2] originally for images, obtained by simply removing the total variation prior, which is a domain specific element of the attack. Moreover, if we examine the state of the art optimization-based attacks for the image [3, 4, 6] and the text domains [5] and strip away all domain specific elements that are not applicable for the tabular setting, we remain with either the squared error or the cosine similarity as the optimization objective. As the cosine similarity tends to outperform the squared error ([2, 5] and also confirmed by our experiments, see the table below), we chose to employ it as the strongest baseline that relies on the state of the art prior work on other domains.
>
> Reconstruction error on the Adult dataset.
> | batch size | Cosine Baseline         | Squared Error Baseline            |
> |:-----------|:-----------------:|:-------------------------:|
> | 8          |**96.1** $\pm$ **5.2**       |64.1 $\pm$ 2.7             |
> |16          |**79.3** $\pm$ **4.4**        |63.1 $\pm$ 2.1             |
> |32          |**69.7** $\pm$ **2.2**        |63.4 $\pm$ 1.4             |
>
> If you have further suggestions for existing baselines to compare against, we are glad to take them into account in our experimental evaluation.
>
> **Q4: Could the mixed-integer problem be tackled directly, without continuous relaxations, with methods such as Genetic Algorithms or Branch-and-Bound?**
>
> No, we do not believe this would be possible with the same efficiency and scalability as TabLeak; with the lack of such methods applied to data leakage problems being also telling of their considered promise. It is highly non-trivial to make mixed discrete-continuous optimization work in this setting, as the search space is exponential in the batch size and the gradient is not linear in the input data (due to the classification layer). A quick back of the envelope calculation shows that on the Adult dataset, with 8 discrete and 6 continuous features, at batch size $n$, the discrete features would reside in a search space of approximately the size of $10^{7.5 n}$. Additionally, in a naive implementation of BaB, the branching could be done by solving the continuous part of the optimization problem, which means a full execution of an optimization based gradient leakage attack (order of minutes in execution time). Clearly, such a naive approach could result in prohibitive time complexity, and as such, we believe that effectively applying methods like BaB to this problem is hardly feasible.
>
> **Q5: Could a better method than ensembling be used to achieve variance reduction?**
>
> We chose pooling, as (i) it requires no further assumptions (e.g., assumptions about the data distribution), (ii) is an easy but effective method for improving reconstruction results on tabular data, as shown in Table 1 and in Appendix E.1 (added as part of the response to Reviewer $\Rtr$), and (iii) it allows for a practical way of measuring the uncertainty in the reconstructed features (3.3 Entropy-based Uncertainty Estimation), which is a key element of our method. We would be glad to explore further variance reduction schemes if you can point us to some better methods, elaborating also on their advantages.
>
> **Q6: “There is work in distributed computing on how to effectively explore parallelization.”**
>
> This sounds interesting, could you please provide us with concrete pointers to such work, so we could explore them in the context of our work?

---

> ### Author Response · Authors · 2022-11-14
> **Response Part 1/3**
>
> Thank you for your time spent reviewing our paper and providing helpful feedback. Further, thank you for acknowledging our contribution of identifying the unique aspects of data leakage on tabular data; which we believe is an important feat for any inaugural work in a new setting. We also highly appreciate your acknowledgement of the good and easy-to-follow writing style, correctness, and the reproducibility of the paper.
>
> We address the concerns raised in your review below:
>
> **Q1: Is the novelty and significance of the work low due to the simplicity of the proposed method?**
>
> No, we believe that our paper offers sufficient novelty in an important problem. The main sources of the novelty in our work are the identification of the unique challenges in tabular data leakage (as also stated by you), which is an interesting problem (Reviewer $\Ro$), offering a simple yet very strong approach to tackling them (appreciated by Reviewer $\Rtr$ both in its simplicity and results), and conducting an extensive experimental evaluation in understanding the pitfalls of the privacy of tabular federated learning systems (acknowledged also by Reviewer $\Rtr$, and extended by an investigation of the increased variance challenge of mixed-type tabular data in Appendix E.1). For a more thorough discussion on this, please see our meta reply to all Reviewers.
>
> **Q2: Does the Gumbel-Softmax provide a better continuous relaxation than softmax for reconstructing categorical features from gradients?**
>
> No, because the underlying theoretical optimization problem in data leakage is non-stochastic. The Gumbel-Softmax provides a differentiable method to sample from categorical distributions, i.e., when using continuous optimization over discrete random variables [1]. However, in the optimization problem for data leakage, the discrete features have no random nature, and hence the sampling introduced by the Gumbel-Softmax is not necessary. In fact, it is detrimental to the performance of the attack, as demonstrated by our experiments below, where we compare TabLeak (no pooling) to TabLeak (no pooling) with the softmax replaced by the Gumbel-Softmax:
>
> | batch size | TabLeak (no pooling)     | Gumbel (cool)             | Gumbel (constant)             |
> |:-----------|:----------------------------:|:-------------------------:|:-----------------------------:|
> | 8          |**93.9** $\pm$ **10.2**       |39.1 $\pm$ 5.1             |46.2 $\pm$ 5.7                 |
> |16          |**84.5** $\pm$ **9.9**        |41.6 $\pm$ 4.5             |49.4 $\pm$ 4.9                 |
> |32          |**72.4** $\pm$ **4.6**        |42.0 $\pm$ 3.5             |49.4 $\pm$ 4.0                 |
>
> Details: The table presents the reconstruction accuracy on the Adult dataset when attacking FedSGD with TabLeak (no pooling), Gumbel (cool), and Gumbel (constant). Gumbel (cool) and Gumbel (constant) are instantiations of TabLeak (no pooling) with the softmax replaced by the Gumbel-Softmax, with the former using an exponential temperature cooling (cool) and the latter having a constant temperature (constant).

---

> ### Author Response · Authors · 2022-11-25
> **Follow-up**
>
> We thank you once again for your feedback, and would like to ask you to consider reassessing our paper in light of our rebuttal. Please, reach out to us with any remaining concerns and questions; we are more than happy to continue the discussion.

---

### Official Review · Reviewer_oUHx · 2022-11-04

**Confidence:** 3
**Correctness:** 3
**Technical Novelty And Significance:** 2
**Empirical Novelty And Significance:** 2
**Recommendation:** 3

**Clarity, Quality, Novelty And Reproducibility:**

This paper is not well written and it is hard to understand its content. Also, the methods in this paper are quite standard and do not show much novelty.

**Strength And Weaknesses:**

Strength:
This paper considers a very interesting topic.

Weakness:

First, the solutions to tackle the challenge brought by tabular data are quite standard.

Second, this paper is not well written. Certain sentences are very obscure and it is hard to understand what the authors are trying to say.
Also, the grammars are not correct in certain sentences. One example is the sentence following Equation (2).
Moreover, in this paragraph, the tense should be consistent. The present tense is used in the beginning of the paragraph, however is switched to past tense for unknown reasons. Although this might be correct in grammar, it is very annoying to the readers.

**Summary Of The Paper:**

This paper considers the data leakage attack in federated learning and focuses on the tabular data. A new method called TabLeak is proposed, which consists of three ingradients: (Section 3.1) softmax structural prios; (Section 3.2) pooled ensembling; and (Section 3.3) entropy-based uncertainty estimation. The combined attack is given in Section 3.4. Numerical experiments is presented in Section 4 and conclusions are drawn in Section 5.

**Summary Of The Review:**

Please see my comments above. Although this paper may contain some intellectual merit, the poor presentation makes it hard for readers to appreciate it.

---

> ### Author Response · Authors · 2022-11-14
> **Response**
>
> $\newcommand{Ro}{\textcolor{purple}{oUHx}}$
> $\newcommand{Rt}{\textcolor{green}{x5xg}}$
> $\newcommand{Rtr}{\textcolor{blue}{Gtmh}}$
> Thank you for your time and valuable input. We are very pleased to learn that you share our view, and also consider data leakage attacks on federated learning systems with tabular data a very interesting problem. As you have mentioned that parts of the paper remain unclear to you, we are happy to answer your questions, if you are ready to provide us with more detail.
>
> Further, below we address your concerns:
>
> **Q1: The proposed solution is quite standard**
>
> Even though the proposed algorithm is built from simple elements, we believe this is not a sufficient reason to dismiss the contribution of our work. Even more so, the simplicity of the approach can be viewed as a key strength, as stated by Reviewer $\Rtr$; especially in light of our strong results. We believe that we provide significant contributions to this important and interesting problem (as also expressed by the Reviewer) by, as first, identifying the key challenges and unique aspects of data leakage in the tabular setting (as mentioned by Reviewer $\Rt$), providing a strong first solution to these challenges in TabLeak (as acknowledged and appreciated by Reviewer $\Rtr$), and by thoroughly exploring the privacy pitfalls of the important problem of tabular federated learning through our extensive experiments, as praised by Reviewer $\Rtr$. For a more thorough discussion of this point, please see our meta response to all Reviewers.
>
> **Q2: Writing**
>
> Despite Reviewers $\Rt$ and $\Rtr$ pointing out that the paper is well written and easy to follow, we take all criticism seriously, and appreciate your feedback. As such, we have updated the paragraph mentioned in your review for easier readability. If you have further concrete suggestions, we are happy to address them, increasing the quality of our paper.

---

> ### Author Response · Authors · 2022-11-25
> **Follow-up**
>
> Once again, we would like to thank you for your review, and would like to ask you to consider reassessing our work post-rebuttal. If you have any comments, concerns, and question left, or newly arising, we are happy to engage in a discussion addressing them.

---

### Author Response · Authors · 2022-11-14
**General Response**

$\newcommand{Ro}{\textcolor{purple}{oUHx}}$
$\newcommand{Rt}{\textcolor{green}{x5xg}}$
$\newcommand{Rtr}{\textcolor{blue}{Gtmh}}$
We would like to thank all Reviewers for their time spent reviewing our paper and their valuable feedback. Based on the received comments, we have made the following changes to the paper: (i) added chapter E to the Appendix investigating the pooling component of our method in detail, i.e., studying the increased variance challenge and the effectiveness of pooling mitigating it (E.1), the impact of the ensemble size (E.2), and the choice of mean vs median pooling (E.3); (ii) extending the experiments on the varying model sizes in Appendix C.2; (iii) clarifying several points in the main body of the paper, based on the comments of Reviewers $\Ro$ and $\Rtr$. Further, below we summarize a common concern of Reviewers $\Ro$ and $\Rt$ and address it in this joint answer. We address the other comments, questions, and concerns of the reviewers in individual responses.

**Q1: Does the technical simplicity of the method limit the novelty and significance of the paper?**

No, we believe that our work considers an important problem, as signified also by the PETs challenge (https://petsprizechallenges.com/) and the abundance of industrial applications of tabular data, especially in privacy critical settings, such as finance or healthcare [1]. Further, we make important and novel contributions in: (i) identifying the unique aspects of data leakage in the tabular setting (as acknowledged by Reviewer $\Rt$), (ii) proposing a simple yet very effective solution to them (praised by Reviewer $\Rtr$ as the main strength of the paper), achieving strong results, significantly outperforming the baseline attack transferred from the image domain (e.g., recovering 77.6\% of the data correctly at batch size of 32 on Adult, Reviewer $\Rtr$ also acknowledging our strong results), (iii) providing an extensive experimental evaluation of the method, being among the first to consider practically highly relevant settings such as attacks against federated averaging, and exposing their fundamental vulnerability. Altogether, our work serves as an important addition to the federated learning data leakage literature, extending the research onto the industry-relevant and privacy-critical domain of tabular data, distilling the problem to its key challenges, and proposing a strong and reproducible (as recognised by all Reviewers) baseline for future research.

**References**

[1] Borisov, Vadim, et al., "Deep neural networks and tabular data: A survey." arXiv preprint arXiv:2110.01889. 2021.

---

### Decision · Program_Chairs · 2023-01-20

**Decision:**

Reject

**Justification For Why Not Higher Score:**

Reviewers have concerns on the novelty and the significance of the contribution, as the proposed algorithm is the plug-in of several existing techniques. It is also questionable whether the baseline methods are the existing state-of-the-art

**Justification For Why Not Lower Score:**

N/A

**Metareview: Summary, Strengths And Weaknesses:**

In this paper, the authors studied the task of data reconstruction from using the gradients from federated learning, in particular, they attempted to solve an interesting problem of data reconstruction for tabular (categorical) data. The challenge is that the original data are categorical and the reconstruction involves mix-integer optimization. The proposed solution is based on  softmax parametrization and pooling. Their empirical evaluations showed good performance compared to some baseline.

Reviewers have concerns on the novelty and the significance of the contribution, as the proposed algorithm is the plug-in of several existing techniques. It is also questionable whether the baseline methods are the existing state-of-the-art. Reviewers appreciated the diligent detailed rebuttals. Nevertheless, the overall assessment is that the quality of the submission does not reach bar for publication in this venue.